# Unsupervised pretraining in biological neural networks

Lin Zhong[1 ✉], Scott Baptista[1], Rachel Gattoni[1], Jon Arnold[1], Daniel Flickinger[1], Carsen Stringer[1,2] & Marius Pachitariu[1,2 ✉]

Representation learning in neural networks may be implemented with supervised or unsupervised algorithms, distinguished by the availability of instruction. In the sensory cortex, perceptual learning drives neural plasticity[1–13], but it is not known whether this is due to supervised or unsupervised learning. Here we recorded populations of up to 90,000 neurons simultaneously from the primary visual cortex (V1) and higher visual areas (HVAs) while mice learned multiple tasks, as well as during unrewarded exposure to the same stimuli. Similar to previous studies, we found that neural changes in task mice were correlated with their behavioural learning. However, the neural changes were mostly replicated in mice with unrewarded exposure, suggesting that the changes were in fact due to unsupervised learning. The neural plasticity was highest in the medial HVAs and obeyed visual, rather than spatial, learning rules. In task mice only, we found a ramping reward-prediction signal in anterior HVAs, potentially involved in supervised learning. Our neural results predict that unsupervised learning may accelerate subsequent task learning, a prediction that we validated with behavioural experiments.

Many neurons in sensory cortical areas change their responses during learning to encode task stimuli in a more-selective or more-efficient way[1–13]. These changes have been interpreted as a possible basis for perceptual learning. A less common interpretation for neural plasticity is unsupervised learning, which has been proposed in multiple theoretical frameworks to describe how the brain learns from sensory experience, without the need for task labels or task feedback[14–19]. However, experimental evidence for such theories is limited to a few indirect observations. In primates, changes of neural tuning have been observed in the inferotemporal cortex after repeated exposure to temporally linked stimuli, even in the absence of rewards[20,21]. Neural plasticity in the critical period during development depends, in some cases, on exposure to relevant sensory stimuli[22]. In the mouse visual cortex, anticipatory responses to learned stimulus sequences have been interpreted as evidence for predictive coding arising from unsupervised learning[23,24]. Neural plasticity in the hippocampus has sometimes been interpreted as sensory compression[25,26], which is also a type of unsupervised learning, although it has typically been linked to spatial rather than sensory representations[27,28]. Finally, multiple types of synaptic plasticity have been observed that do not require supervision[29,30], although it is not known whether such plasticity occurs in behaving animals and what its effects are on neural representations and behaviour. It is thus still not known how widely unsupervised learning may affect sensory neural representations.

Here we found that most of the neural plasticity in the visual cortex after task learning was replicated in mice with unsupervised exposure to the same visual stimuli. We found the only exception in anterior visual areas, which encoded unique task signals potentially used for supervised learning. Below we describe a sequence of experiments designed to probe the roles of supervised and unsupervised learning across multiple visual computations, before and after learning.

## Supervised and unsupervised plasticity

Similar to previous work[10], we designed a visual discrimination task in head-fixed mice running through linear virtual reality corridors (Fig. 1a). Mice had to discriminate between visual texture patterns in two corridors; these corridors were repeated in pseudo-random order. The visual patterns in each corridor were obtained as 'frozen' crops from large photographs of naturalistic textures. For simplicity, we denote the stimuli as 'leaf' and 'circle', even though other visual stimuli were also used in some mice ('rock' and 'bricks'; see Methods and Extended Data Fig. 1a). The visual stimuli were spatial frequency matched between categories to encourage the use of higher-order visual features in discrimination. A sound cue was presented at a random position inside each corridor and was followed by the availability of water in rewarded trials only (Fig. 1a). After approximately 2 weeks of training (Fig. 1b), mice demonstrated selective licking in the rewarded corridor in anticipation of reward delivery (Fig. 1c,d; error bars on all figures represent s.e.m.). After learning, we introduced unrewarded test stimuli 'leaf2' and 'circle2', which were different frozen crops of the same photographs. We then continued training with unrewarded leaf2 until the mice stopped licking to this stimulus, at which point we introduced another test stimulus ('leaf3') as well as spatially shuffled versions of leaf1 (Fig. 1b).

Mice in the unsupervised cohort also ran through the same corridors for similar periods of time, but did not receive water rewards and were not water restricted. We also studied a cohort of mice that ran through a virtual reality corridor with gratings on the walls. These served as a control to show the effect of simple exposure to virtual reality, independent of the image patterns shown. To ensure that all mice had a comparable visual experience, we fixed the speed of the virtual reality when mice ran faster than a speed threshold, and kept the virtual reality

[1]HHMI Janelia Research Campus, Ashburn, VA, USA. [2]These authors contributed equally: Carsen Stringer, Marius Pachitariu. ✉e-mail: zhongl@janelia.hhmi.org; pachitarium@janelia.hhmi.org

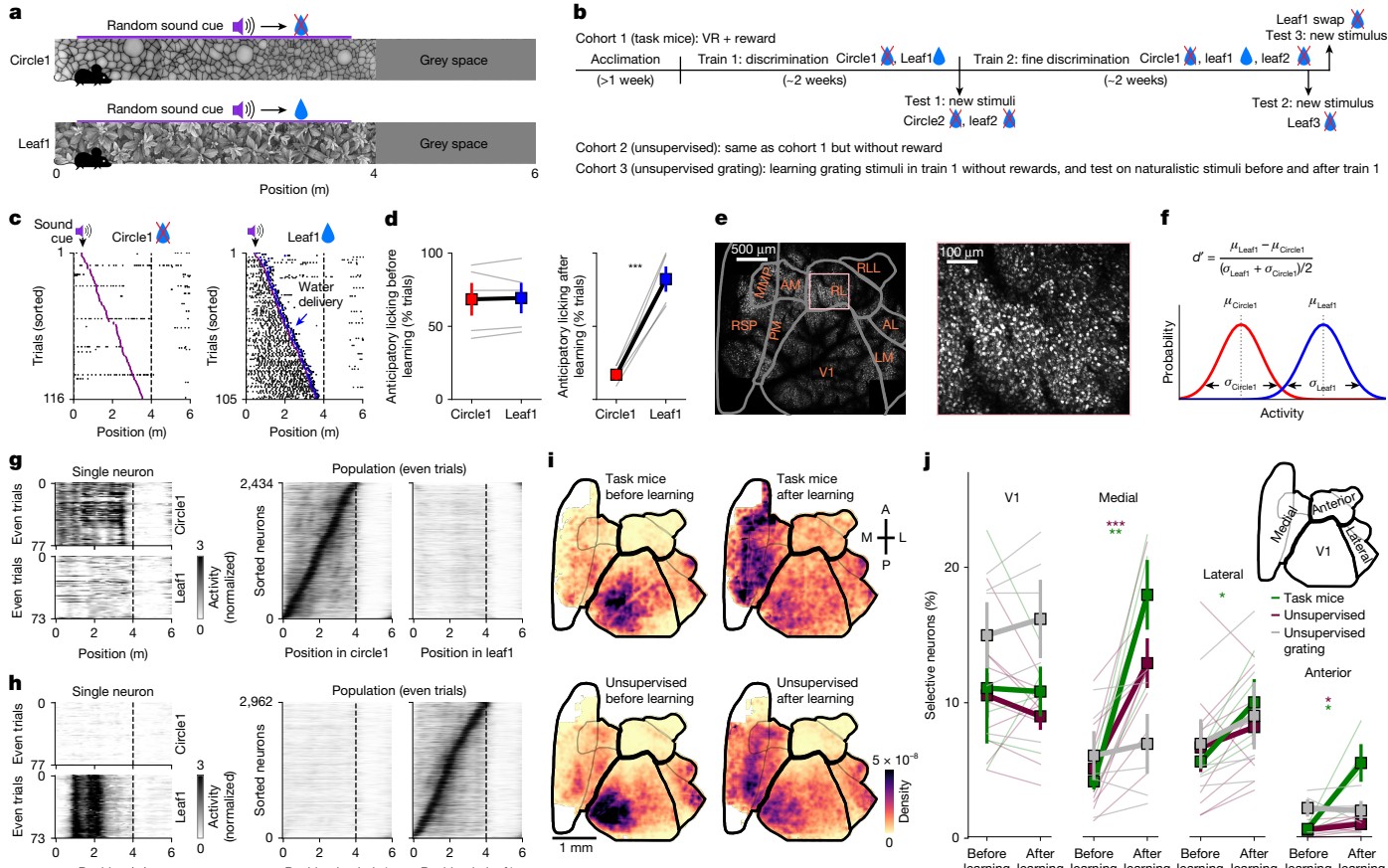

**Fig. 1 | Plasticity in the visual cortex after supervised and unsupervised training. a**, Illustration of the virtual reality (VR) task with a sound cue at a random position in each corridor. Water was available after the sound cue in the rewarded corridor. **b**, Task training timeline in task mice. Mice in the unsupervised cohort experienced the same stimuli or the gratings stimuli without water rewards. **c**, Lick distribution in an example mouse after task learning. Trials were sorted according to the sound cue position. **d**, Performance quantification of anticipatory licks before water was delivered (error bars on all figures represent s.e.m.; the centre values are average; grey lines denote $n = 5$ mice). **e**, Example field of view of the mesoscope (left) and zoomed-in view (right) to illustrate cellular resolution. AL, anterolateral; LM, lateromedial; MMP, mediomedialposterior; RL, rostrolateral; RLL, rostrolaterallateral; RSP, retrosplenial. **f**, Selectivity index $d'$ of neural responses inside the

two corridors. **g**, Single-trial responses of a single circle1-selective neuron as well as the entire population from an example mouse. **h**, Same as panel **g** but for leaf1-selective neurons. **i**, 2D histogram of selective neuron distributions across the field of view, aligned to a map of visual areas. The top and bottom rows are task mice and unsupervised mice, respectively. The left and right columns are before and after learning, respectively. A, anterior; L, lateral; M, medial; P, posterior. **j**, Percentage of neurons with high selectivity in each of the four visual regions defined in the inset ($n = 4$ task mice, $n = 9$ unsupervised mice and $n = 3$ unsupervised with gratings mice (5 sessions); statistical tests in all figures are two-sided Student's $t$-tests, paired or independent as appropriate; see Methods for full details). All data are mean ± s.e.m. *$P < 0.05$, **$P < 0.01$ and ***$P < 0.001$ from paired, two-sided $t$-tests.

stationary otherwise. The overall running speeds were similar before and after learning, and between the task and unsupervised cohorts (Extended Data Fig. 2). We only considered timepoints during running for analysis, which removed time periods when the task mice stopped to collect water rewards.

Before and after learning, we recorded from large neural populations across many visual areas simultaneously using a two-photon mesoscope[31] (Fig. 1e). We ran Suite2p on this data to obtain the activity traces from 20,547 to 89,577 neurons in each recording[32]. For each neuron, we computed a selectivity index $d'$ using the response distributions across trials of each corridor, pooled across positions and for timepoints when the mice were running (Fig. 1f). Neurons with relatively high $d'$ ($d' \geq 0.3$ or $d' \leq -0.3$) responded strongly at some positions inside the leaf1 or circle1 corridor (Fig. 1g,h). To see where these selective neurons were located, we generated 2D histograms of their position in tissue after aligning each session to a pre-calculated atlas of visual cortical areas (Fig. 1i and Extended Data Fig. 1b,c; see also ref. 33). We found that, after learning, many selective neurons emerged in medial visual areas, encompassing regions posteromedial (PM), anteromedial

(AM) and mediomedialanterior (MMA), as well as the lateral part of the retrosplenial cortex[33] (Fig. 1j). This region, which we refer to as the 'medial' visual region, showed similar changes in neural selectivity to natural images for both the task and the unsupervised cohorts (but not for the cohort of mice with exposure to gratings in virtual reality; Extended Data Fig. 1d). The pattern of changes did not depend on the precise threshold used to determine selectivity (Extended Data Fig. 1e), and was also observed separately for neurons tuned to the leaf1 and circle1 stimuli, respectively (Extended Data Fig. 1f). We did not observe changes in the lateral regions, and the anterior regions were only modulated in the supervised condition (see Fig. 4 for more on this). We also observed some plasticity in V1, where the medial part showed a small but significant decrease in selectivity in the unsupervised cohort (Extended Data Fig. 1g), as well as some minor changes of selectivity when separated by stimulus (Extended Data Fig. 1f). However, the overall fraction of selective V1 neurons did not change by much, similar to some previous studies[34,35], but different from other studies[10,13] (see Discussion).

Thus, the distribution of neural plasticity across visual regions mostly did not depend on task feedback or supervision. Note that mice in

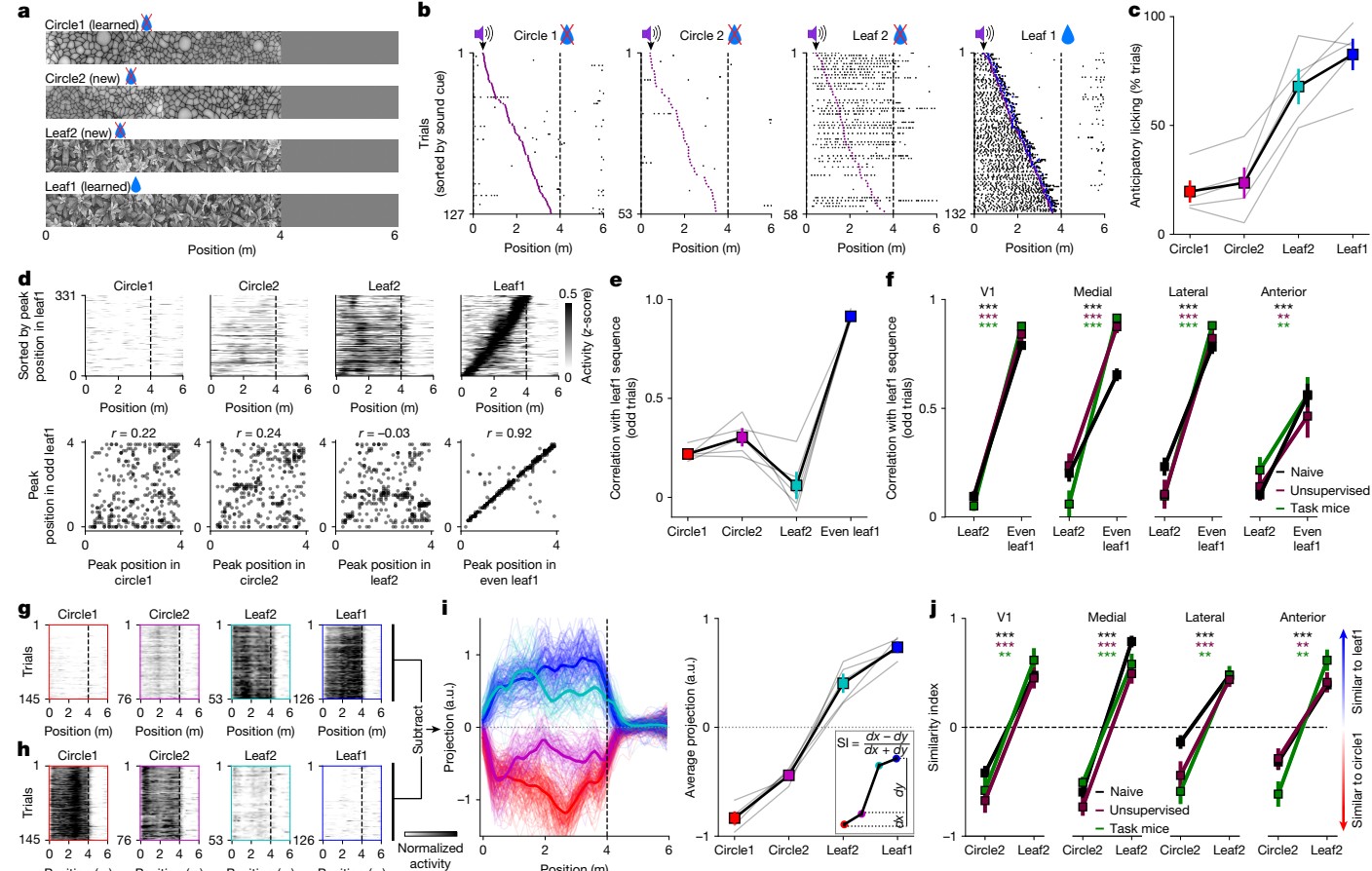

**Fig. 2 | Comparing visual and spatial coding on test stimuli. a**, Stimuli in the test 1 session (see timeline in Fig. 1b). **b**, Lick rasters for an example mouse (blue dots represent reward delivery). **c**, Anticipatory licking behaviour in test 1 (*n* = 5 mice). **d**, Example neural responses from the medial region in a mouse from the task cohort, sorted by preferred position in the leaf1 corridor on held-out trials (top), and scatter plots of preferred positions in the leaf1 corridor versus all other corridors (bottom). **e**, Correlation from panel **d** (bottom) for each mouse in the task cohort (*n* = 5 mice). **f**, Same as panel **e** but only for the leaf 2 and leaf 1 corridors, shown across regions (*n* = 5 task mice;

*n* = 7 unsupervised mice; *n* = 9 naive mice, 11 sessions). **g**,**h**, Example leaf1-selective (**g**) and circle1-selective (**h**) neurons from the medial area shown as a population average for each trial. **i**, Projections onto the coding direction in the medial area, defined as the difference between the leaf1-selective and circle1-selective populations in panels **g**,**h**. Time course (left, one mouse), average over trials (right, five mice) and definition of similarity index for circle2 (inset) are shown. a.u., arbitrary units. **j**, Similarity index for new stimuli of the same mice as in panel **f**. All data are mean ± s.e.m. \*\*P < 0.01 and \*\*\*P < 0.001 from paired, two-sided *t*-tests.

the task cohort also had to learn to obtain water and learn the relation between water, stimuli, corridor positions and the sound cue. They also had different experiences in the rewarding corridor, where they stopped, drank water and restarted running; however, we did not consider these timepoints for analyses. Despite all these differences, neural plasticity was similar to the unsupervised cohort, suggesting that most of the plasticity in these areas relates to the stimuli directly, irrespective of other factors.

### Visual, not spatial, representations

It is possible that the neural plasticity that we observed was due to spatial learning and navigation signals, which have been found to modulate firing rates even in the visual cortex[36]. Alternatively, the neural plasticity might be due to adaptation to the visual statistics of the natural images that we presented[14]. To distinguish between a spatial and a visual plasticity hypothesis, we next introduced two unrewarded test stimuli, leaf2 and circle2, which contained similar visual features to leaf1 and circle1, respectively, but were arranged in different spatial configurations (Fig. 2a). We found that mice only licked to leaf2 and not to circle2, probably due to their visual similarities with the trained stimuli (Fig. 2b,c).

The spatial plasticity hypothesis suggests that neurons would fire in a similar sequence to learned and new exemplars of the same

category. We tested this directly by sorting neurons according to their sequence of firing in the leaf1 corridor. This sorting did not induce similar sequences for the leaf2 corridor in the medial regions (Fig. 2d). The neural sequences in leaf1 and leaf2 were in fact uncorrelated in all regions, suggesting a non-spatial coding scheme (Fig. 2e,f). Furthermore, the sequence correlations did not match the behavioural strategy of the mice (Fig. 2c,e). Similar results were found when analysing the sequences of the circle1-selective neurons (Extended Data Fig. 3a,b).

The visual plasticity hypothesis suggests that the statistics of visual features (that is, 'leafiness') are learned regardless of where the features occur in the corridor. To test this, we designed an analysis that uses the top selective neurons of each familiar corridor to create a coding direction axis[37] (Fig. 2g,h). Projections of neural data on the coding direction were well separated on test trials of the familiar stimuli, and also on trials of the new leaf2 and circle2 stimuli (Fig. 2i and Extended Data Fig. 3c–e). To quantify this separation, we defined a similarity index computed from the projections on the coding axis. The similarity index clearly distinguished between corridors of different visual categories, in all visual regions, in both task and unsupervised mice as well as in naive mice (Fig. 2j). Thus, the coding direction readout of neural activity matched the behavioural strategy of the mice (Fig. 2c,i), suggesting that the brain areas that we considered use visual rather than

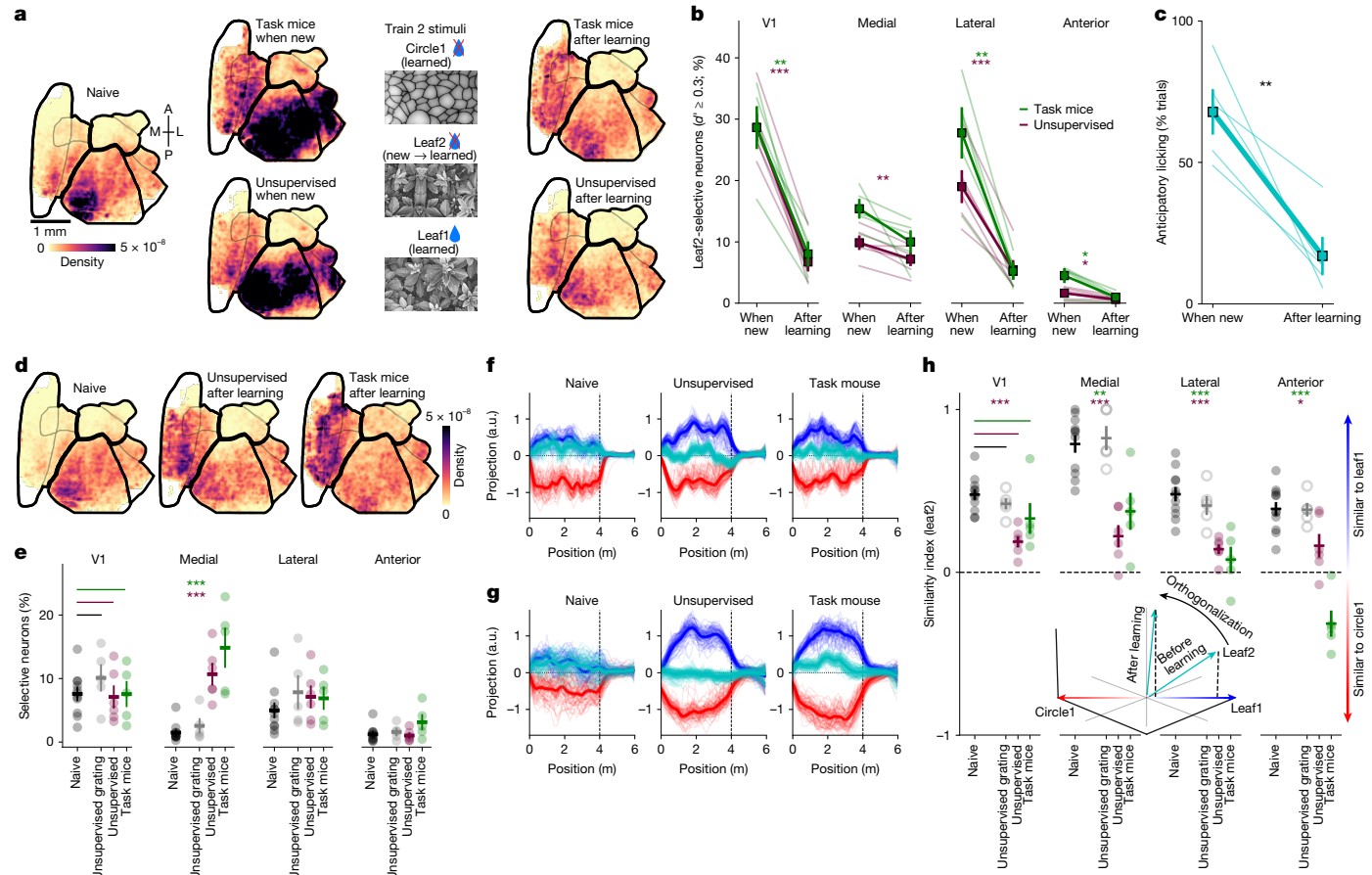

**Fig. 3 | Responses to novel and adapted stimuli and neural orthogonalization.** **a**, Distribution of neurons with $d' \geq 0.3$ between the leaf2 and circle1 corridors in task mice and unsupervised mice, either when new or after learning, as well as in fully naive mice. **b**, Summary of distribution changes in panel **a** across regions ($n = 5$ task mice and $n = 6$ unsupervised mice). **c**, Licking behaviour to leaf2 when new and after learning ($n = 5$ mice). **d**, Distribution of neurons with $d' \geq 0.3$ or $d' \leq -0.3$ between leaf1 and leaf2 in task mice and unsupervised mice after training with leaf2, as well as in fully naive mice. **e**, Summary of panel **d** across regions ($n = 5$ task mice; $n = 6$ unsupervised mice; $n = 9$ naive mice,

11 sessions; $n = 3$ unsupervised with gratings mice, 6 sessions). **f,g**, Example projections on the coding direction between leaf1 and circle1 in V1 (**f**) and the medial region (**g**). **h**, Similarity index (from Fig. 2i) for neural responses to leaf2 (same $n$ as panel **e**). The bottom inset shows a schematic of the orthogonalization effect observed. Neural vectors are referenced with respect to the centre of the leaf1–circle1 axis. All data are mean ± s.e.m. *$P < 0.05$, **$P < 0.01$ and ***$P < 0.001$ from paired, two-sided $t$-tests, except panels **e,h**, which were independent two-sided $t$-tests.

spatial coding. However, the coding direction readout did not directly reflect licking behaviour because it stayed unchanged between trials with early cues, which contained only a few anticipatory licks, and trials with late cues, which contained many licks (Extended Data Fig. 4a–c). An exception to this pattern was, again, the anterior region on leaf1 trials (see below for the likely reason).

## Novelty and orthogonalization

Although the neurons selective to familiar stimuli also responded to the new stimuli, they were by far not the most responsive neurons to these new stimuli. Selecting neurons by $d'$ between leaf2 and circle1, we found a large population of leaf2-selective neurons in V1 and the lateral visual areas (Fig. 3a; see also Extended Data Fig. 5a for circle2-selective neurons). The number of leaf2-selective neurons decreased substantially after an additional week of training with the leaf2 stimulus, suggesting adaptation (Fig. 3a,b). By contrast, the number of leaf1-selective neurons showed little or no change after this additional learning phase (Extended Data Fig. 5b). Thus, V1 and lateral HVAs appear to be responding to the stimulus novelty, as reported by some previous studies in V1 (ref. 38), and the responsive neurons were similarly distributed across brain areas in the task and unsupervised cohorts.

The task mice eventually stopped licking to leaf2 (Fig. 3c), thus showing strong discrimination between leaf1 (rewarded) and leaf2

(unrewarded). We hypothesized that this behaviour may be accompanied by changes in neural discrimination, similar to the changes after mice learned the distinction between leaf1 and circle1 (Fig. 1j). To test this, we selected neurons based on their $d'$ between leaf1 and leaf2, and compared the fraction of tuned neurons across areas in naive mice as well as in the task and unsupervised cohorts. Again, we observed a substantial increase in selectivity in the medial HVAs, in both the task and the unsupervised cohorts, but not in the group exposed to the virtual reality gratings (Fig. 3d,e).

Because leaf1 and leaf2 neural representations were similar in naive mice (Fig. 2j), we hypothesized that the fine behavioural discrimination between leaf1 and leaf2 stimuli requires orthogonalization of their neural representations[28,35]. We tested this by comparing the projections of leaf2 onto the leaf1–circle1 coding direction (Fig. 3f,g). Compared with naive mice, this projection was reduced after both supervised and unsupervised training, across all visual regions but most strongly in the medial HVAs (Fig. 3h). There was no change in the control group trained on the virtual reality gratings compared with naive mice. Thus, the responses to leaf2 became orthogonal to the leaf1–circle1 axis, even in the unsupervised mice in which leaf1 and leaf2 had the same valence. Together, these observations describe the complex dynamics of the representation of a new stimulus (leaf2) as it becomes familiar, in both supervised and unsupervised conditions.

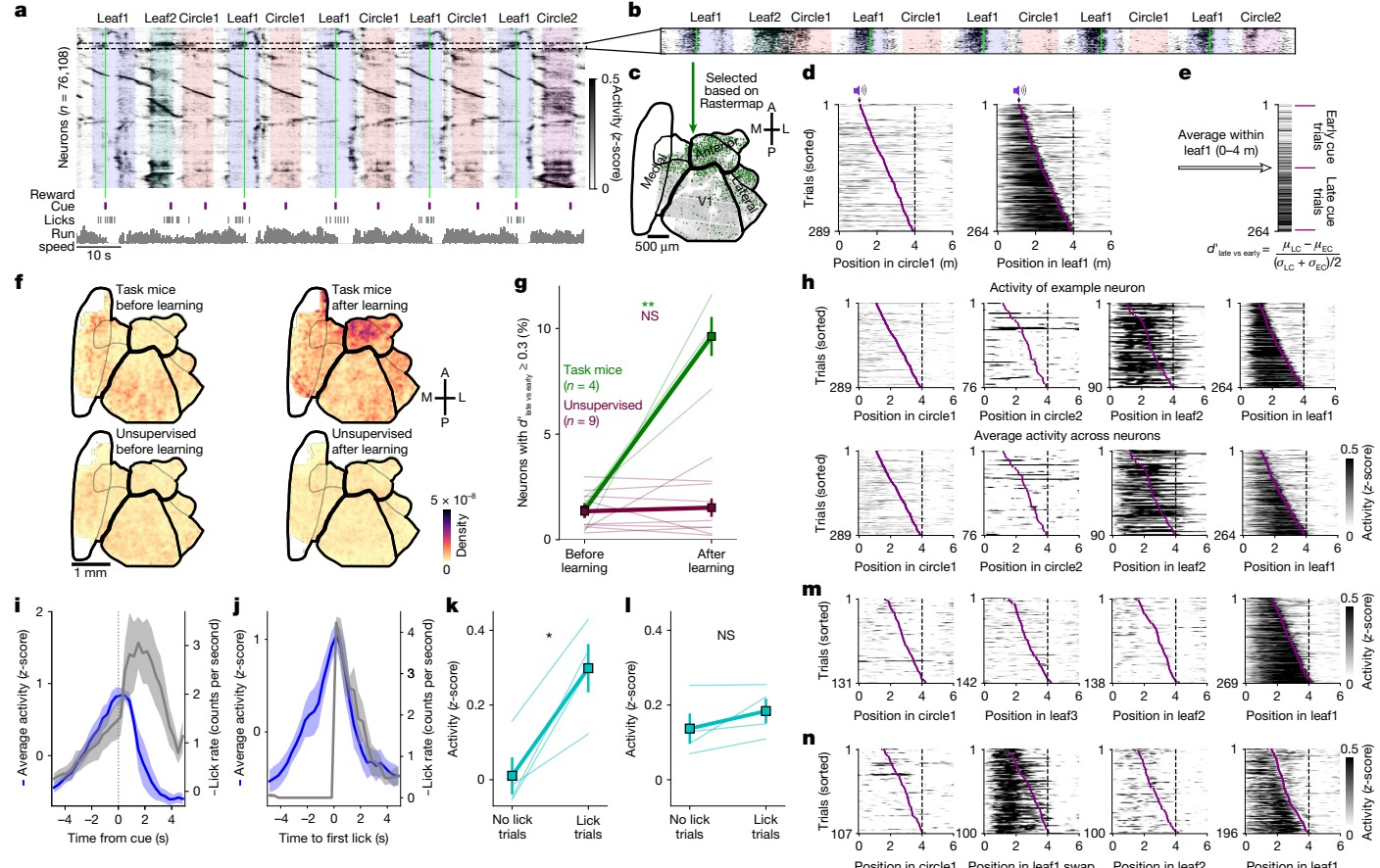

**Fig. 4 | A reward-prediction signal in supervised training only. a**, Raster plot of simultaneously recorded neurons, over a period of approximately 2.5 min, with behavioural annotations. Rastermap was used to sort neurons across the *y* axis. **b**, Zoomed-in view of a vertical segment in Rastermap that corresponds to a group of neurons active in the leaf1 corridor. **c**, Spatial distribution of selected neurons. **d**, Population average response for the neurons selected in panel **b** across trials of circle1 and leaf1, sorted by sound cue position. **e**, A discrimination index ($d'_{\text{late vs early}}$) was constructed to compare trial-averaged neuron responses between trials with early versus late rewards. EC, early cue; LC, late cue. **f**, Distribution of reward-prediction neurons ($d'_{\text{late vs early}} \geq 0.3$) in task and unsupervised mice, before and after learning. **g**, Percentage of reward-prediction neurons in the anterior region.

**h**, Responses of a single selective neuron (top) or the entire population (bottom) across trials in the test 1 session. **i**, Reward-prediction neuron activity and lick rates in the leaf1 corridor, aligned to the sound cue in the test 1 session, averaged across mice (*n* = 5). Dashed line indicates 0. **j**, Same as panel **i** but aligned to the first lick in the corridor for trials in which the first lick was after 2 m (*n* = 4 mice). **k**, Average activity of reward-prediction neurons inside the leaf2 corridor on trials with or without licks in the test 1 session (*n* = 4 mice). **l**, Same as panel **k** but for the leaf1-selective neurons in the medial regions (defined as in Fig. 1h). **m**,**n**, Same as panel **h** (bottom) for test 2 (**m**) and test 3 (**n**) sessions. For panels **d**, **h**, **m** and **n**, the dashed line indicates end of the corridor. All data are mean ± s.e.m. *\*P* < 0.05, *\*\*P* < 0.01 and not significant (NS) from paired, two-sided *t*-tests.

## Visual recognition memory

By this stage in the training, mice had been exposed to the leaf1 stimulus for approximately 4 weeks, and the task mice had learned to distinguish it from leaf2. We hypothesized that a more detailed representation of leaf1 emerged to support the visual recognition memory of this stimulus. To test this, we first introduced a new exemplar of the leaf category ('leaf3'; Extended Data Fig. 6a). Mice withheld licking to this stimulus, similar to their behaviour on the unrewarded, trained leaf2 stimulus (Extended Data Fig. 6b). This behavioural choice mirrored a change in neural tuning properties: the neural projections of leaf3 trials onto the coding axis of leaf1–leaf2 were strongly biased towards the leaf2 direction (Extended Data Fig. 6c). This asymmetry was present in both the task and the unsupervised cohorts, and it was not present in naive mice (Extended Data Fig. 6d,e). All visual regions behaved in this manner, except the anterior region where the coding asymmetry was more pronounced for supervised than unsupervised training, and the representation of the 'control' stimulus (circle1) was also biased towards the leaf2 axis. As we see in the next section, this may be due to the reward valence coding in anterior HVAs.

To further test the visual recognition memory for the leaf1 corridor, we introduced two new corridors with spatially swapped portions of leaf1 (Extended Data Figs. 6f and 7a). We reasoned that these new corridors should disrupt the mice if they had memorized only the beginning of the corridor, or if they had used a purely spatial-based memorization strategy. We found no such disruption, with the mice licking in the swapped corridors at levels comparable with the leaf1 corridor (Extended Data Fig. 6g). Thus, the swapped corridors were still recognized as visually similar to the rewarded leaf1 corridor, as opposed to being recognized as a different exemplar of the leaf category (such as leaf2 or leaf3). This behaviour was supported by a neural coding strategy that tied neural response vectors to their respective visual stimulus locations (Extended Data Figs. 6h and 7b–d). This visual encoding was present in all areas in both the supervised and the unsupervised training conditions, and it was also present before training in V1, as expected for a purely visual representation (Extended Data Fig. 6i). The distribution and position of first licks in the swapped and original corridors were not significantly different, suggesting that mice were not using the early portion of the corridor to do the discrimination (Extended Data Fig. 7e–g). The medial region did not appear to strongly

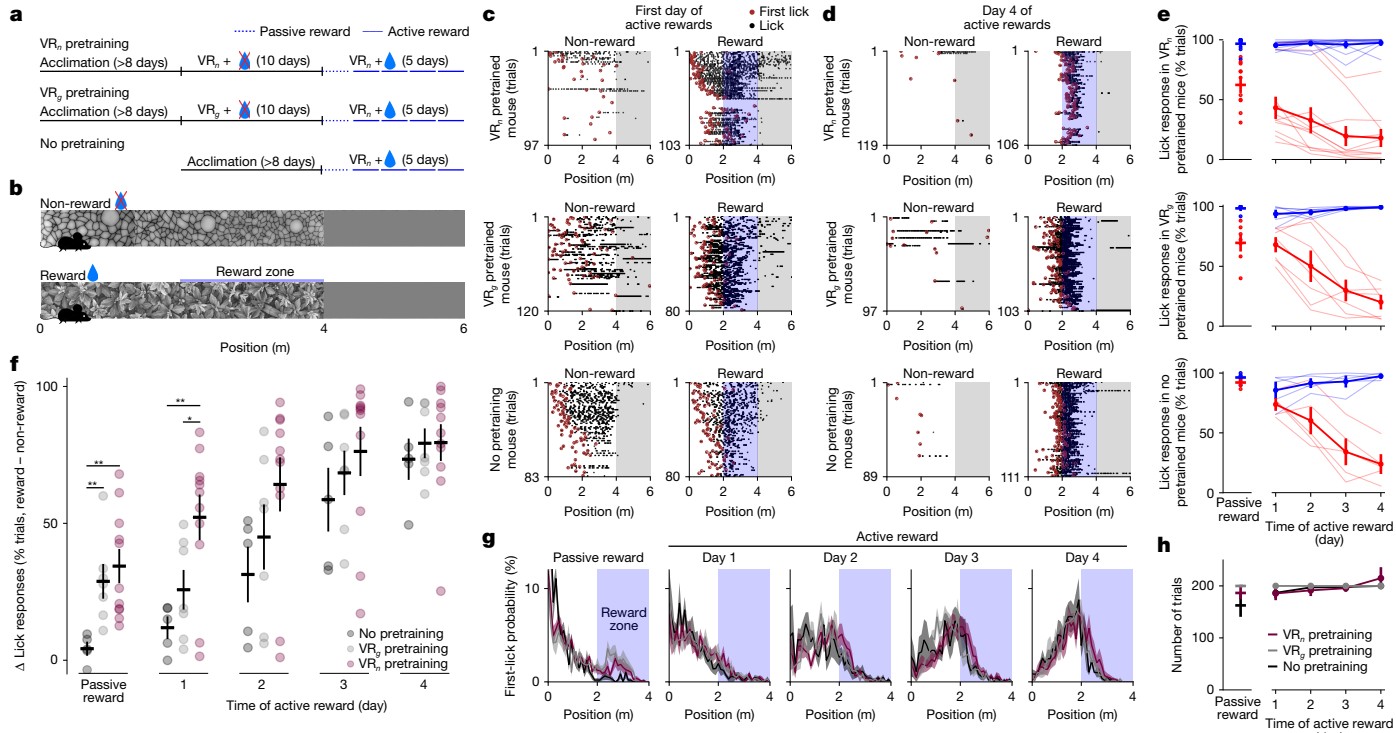

**Fig. 5 | Unsupervised pretraining accelerates subsequent task learning.**
**a**, We trained three new cohorts of mice with or without a pretraining step of running through the virtual reality corridor without rewards for 10 days. All cohorts were trained in the task for 5 days. $VR_g$, virtual reality gratings; $VR_n$, virtual reality naturalistic stimuli. **b**, Task structure (sound cue was removed and rewards were delivered deterministically in the second half of the reward corridor). Mice had to lick to obtain the reward, except on the first day when rewards were delivered passively at the end of the corridor. Each mouse was trained on a separate combination of wall textures from among four stimuli (Extended Data Fig. 1a). **c**, Licking of example mice from each cohort on the first day of training with active rewards (grey background indicates grey corridor). **d**, Same as panel **c** but for the last day of training. **e**, Average lick responses across days for each cohort of mice ($n = 11$ $VR_n$ pretrained mice, $n = 7$ $VR_g$ pretrained mice and $n = 5$ no pretrained mice). **f**, Performance summary (difference in lick responses) across days for each cohort in panel **e**. **g**, Distribution of first licks across days for each cohort in panel **e**. **h**, Number of trials per day for each cohort in panel **e**. All data are mean ± s.e.m. *$P < 0.05$ and **$P < 0.01$ from two-sided $t$-tests.

encode a sequence before learning (Extended Data Fig. 6h,i), but this result may have been due to lower signal-to-noise ratio from an overall weaker stimulus tuning before learning (Fig. 1j).

Thus, for both the leaf3 and the leaf1-swap stimuli, behavioural responses were linked to the patterns of neural responses during supervised training, but these patterns also emerged after unsupervised training.

### Reward prediction in anterior HVAs

Having found multiple similarities between the supervised and unsupervised conditions, we next asked whether a more targeted analysis could reveal differences. As we did not know in advance what to look for, we used Rastermap, a visualization method for large-scale neural responses[39]. Rastermap reorders neurons across the $y$ axis of a raster plot, so that nearby neurons have similar activity patterns. Inspected in relation to task events, Rastermap can reveal single-trial sequences of neural activity tied to corridor progression, as well as other signals that may be related to task events such as rewards and sound cues (Fig. 4a). One of the signals that we found with Rastermap corresponded to a neuronal cluster that turned on specifically in the leaf1 corridor but not in the circle1 corridor, and was turned off by the delivery of reward (Fig. 4a,b). These neurons were distributed in the anterior HVAs (Fig. 4c), and their activity was strongly suppressed by the delivery of reward (Fig. 4d), similar to a reward-prediction signal[40,41].

Having found a putative task-related population with Rastermap, we next quantified the task modulation at the single-neuron level across mice. For this, we developed an index $d'_{\text{late vs early}}$, which compares neural activity on trials where the reward is delivered early with trials where

it is delivered late (Fig. 4e). This index captures differences in anticipatory neural activity, which is more temporally extended on late-reward trials. Other aspects of the neural responses (such as reward-related activity and the stimulus drive) should not affect this index, because they are exactly matched between early-reward and late-reward trials. Selective neurons ($d'_{\text{late vs early}} \geq 0.3$) were distributed primarily in anterior areas in task mice after training (Fig. 4f,g and Extended Data Fig. 8a). We did not observe a similar population when selecting neurons based on circle1 trials with a similar process (Extended Data Fig. 8b,c). Some single neurons in the anterior area with high $d'_{\text{late vs early}}$ encoded the reward-prediction signal robustly at the single-trial level, and their response generalized to the new exemplar leaf2 of the rewarded visual category, similar to the population average of selective neurons (Fig. 4h).

One possibility is that the reward-prediction signal directly reflects licking behaviour, because it was present only in the task mice. We do not think this was the case, because: (1) the reward-prediction signal was strongly suppressed after the cue, whereas licking increased dramatically (Fig. 4i and Extended Data Fig. 8d–f); and (2) the reward-prediction signal started ramping several seconds before the first lick (Fig. 4j). These dynamics are indicative of a reward expectation, especially because the neural prediction signal was higher on leaf2 (unrewarded) trials in which the mouse licked compared with trials in which it did not (Fig. 4k). This distinction was only found in anterior HVAs, and not, for example, in the medial population that we described earlier, or in V1 or in the lateral region (Fig. 4l and Extended Data Fig. 4d).

The reward-prediction signal continued to follow the dynamics of the behaviour itself over the course of training. After training with

leaf2, the reward-prediction signal was suppressed during the leaf2 corridor, and was also absent in the new leaf3 corridor (Fig. 4m). Finally, the reward-prediction signal was present in the swapped leaf1 corridor, again indicating that this signal correlates with the expectation of reward (Fig. 4n).

## Faster task learning after pretraining

Next we tested the potential function of the neural plasticity after unsupervised training. We hypothesized that this plasticity might allow animals to learn a subsequent task faster, similar to how unsupervised pretraining helps artificial neural networks to learn supervised tasks faster, and similar to previous maze learning experiments[42] and home cage exposure experiments[43]. We thus ran a behavioural study in which one cohort of mice ('no pretraining') was trained similarly to the task mice above, whereas a second cohort ('unsupervised pretraining') first underwent 10 days of virtual reality running without rewards (Fig. 5a and Extended Data Fig. 1a). We also added a third cohort of mice similar to the second cohort, except the pretraining stimuli used in virtual reality were gratings instead of natural textures. Compared with our original task, we simplified reward learning by restricting reward delivery to the second half of the reward corridor (the 'reward zone') and removing the sound cue (Fig. 5b). We also included a day of initial training with passive rewards ('day 0') to encourage all mice to start learning.

Mice with unsupervised pretraining of naturalistic textures generally learned the task much faster. For example, one mouse started the first day of task training by licking indiscriminately in both corridors, but stopped licking in the non-reward corridor after approximately 10 trials (Fig. 5c). By the fifth day of training, this mouse was selectively licking only at the beginning of the reward zone (Fig. 5d). By contrast, mice without pretraining or with pretraining on grating corridors did not learn to distinguish the two corridors on the first day of active training (Fig. 5c,d). After 5 days, all three cohorts reached a high discrimination performance, but the unsupervised pretrained cohort learned faster (Fig. 5e,f). Furthermore, most of the learning improvements happened within session (Extended Data Fig. 9). The improved discrimination ability of the pretrained mice was not due to differences in behaviour during task learning: all three cohorts licked at similar positions in the two corridors and ran a similar number of trials (Fig. 5g,h).

## Discussion

Here we showed that unsupervised pretraining has a substantial effect on neural representations in cortical visual areas, and helps mice to learn a supervised task faster. The main region for unsupervised plasticity may be the medial HVAs, as these areas contained emergent representations that strongly discriminated the learned stimuli and emerged with or without task training (Fig. 1ij). Nonetheless, all visual regions showed some changes in tuning after learning (Fig. 3h), even when the number of selective neurons did not increase (Fig. 3e). The medial population did not represent trial-to-trial variability in mouse decision-making, whereas the anterior HVAs did (Fig. 4kl). The reward-prediction signal in anterior HVAs may thus be required for supervised or reinforcement learning (Fig. 4j). Finally, stimulus novelty was represented primarily in the responses of neurons in V1 and lateral HVAs (Fig. 3a,b).

Our results can be related to other reports of neural plasticity in sensory cortices. In the visual cortex, learning can result in more neurons that discriminate the learned stimuli[1,10,13], neurons that discriminate the stimuli better[2,9,12] or neurons that respond more to the learned stimuli[3,6,8]; learning can also add a context dependence to the visual tuning of neurons[36], and it can orthogonalize stimulus representations in V1 (ref. 35). Such changes have typically been interpreted as consequences of task learning because they correlate well with task performance. However, our results suggest that these changes could

have also happened without task training. In V1 specifically, many studies including ours have shown changes in selectivity and tuning, but some studies have also shown an increase in the number of selective neurons[10,13], whereas other studies, including ours, have not[34,35]. This discrepancy might be attributed to differences in the tasks or in how stimulus responses are measured.

Our results can also be compared with those in the hippocampus. Although we have shown that the cortical representations are visual, rather than spatial, in nature, it is possible that hippocampal representations also inherit some visual properties from their inputs[44–46]. Further distinguishing between spatial and visual learning in the same circuits could be a promising direction of future research. Another promising direction would be to find physiological substrates for the changes that we observed, such as synaptic plasticity, and relate those to classical learning rules[29,30] or newer rules, such as the behaviour timescale synaptic plasticity rule[47]. Yet, another future direction could be to relate the unsupervised plasticity to classical theories and models of unsupervised learning[14–18], as well as to modern approaches such as self-supervised learning[19,48–51].

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

## Methods

All experimental procedures were conducted according to the Institutional Animal Care and Use Committee (IACUC), and received ethical approval from the IACUC board at the HHMI Janelia Research Campus.

### Experimental methods

**Animals.** We performed 89 recordings in 19 mice bred to express GCaMP6s in excitatory neurons: TetO-GCaMP6s × camK2a-tTa mice (available as RRID:IMSR_JAX:024742 and RRID:IMSR_JAX:003010)[52]. Of these mice, 13 were male and 6 were female, and ranged from 2 to 11 months of age. Mice were housed in reverse light cycle, and were pair housed with their siblings before and after surgery. The mice had a running wheel in their cage, as well as corncob bedding with Nestlets. During training and imaging periods, we replaced the running wheel with a tube, to potentially motivate the mice to run longer while head fixed. Owing to the stability of the cranial window surgery, we often used the same mice for multiple experiments in the laboratory: two of the mice were used in ref. 53. Two of the mice were raised in complete darkness. We did not see differences compared with normal-reared mice, so we pooled them together. It was not possible to blind the experimenter with respect to behavioural experiments that involved water deprivation (see below). The number of mice in each experimental group was chosen to be comparable with other studies involving complex behaviour and imaging procedures such as ours. Mice were randomly assigned into groups where relevant.

We also used 23 C57 female mice for behaviour-only experiments. These mice were only implanted with a headbar and not a cranial window.

**Surgical procedures.** Surgeries were performed in adult mice (post-natal day 35 (P35)–P333) following procedures previously described[54]. In brief, mice were anaesthetized with isoflurane while a craniotomy was performed. Marcaine (no more than 8 mg kg$^{-1}$) was injected subcutaneously beneath the incision area, and warmed fluids + 5% dextrose and 0.1 mg kg$^{-1}$ buprenorphine (systemic analgesic) were administered subcutaneously along with 2 mg kg$^{-1}$ dexamethasone via the intramuscular route. For mice with cranial windows, measurements were taken to determine bregma-lambda distance and the location of a 4-mm circular window over the V1 cortex, as far lateral and caudal as possible without compromising the stability of the implant. A 4 + 5-mm double window was placed into the craniotomy so that the 4-mm window replaced the previously removed bone piece and the 5-mm window lay over the edge of the bone. After surgery, 5 mg kg$^{-1}$ ketoprofen was administered subcutaneously and the mice were allowed to recover on heat. The mice were monitored for pain or distress, and 5 mg kg$^{-1}$ ketoprofen was administered for 2 days following surgery.

**Imaging acquisition.** We used a custom-built two-photon mesoscope[31] to record neural activity, and ScanImage[55] for data acquisition. We used a custom online $z$-correction module (now in ScanImage), to correct for $z$ and $xy$ drift online during the recording. As previously described[54], we used an upgrade of the mesoscope that allowed us to approximately double the number of recorded neurons using temporal multiplexing[56].

The mice were free to run on an air-floating ball. Mice were acclimatized to running on the ball for several sessions before training and imaging.

**Visual stimuli.** We showed virtual reality corridors to the mice on three perpendicular LED tablet screens, which surrounded each mouse (covering 270° of their visual field of view). To present the stimuli, we used PsychToolbox-3 in MATLAB[57]. The virtual reality corridors were each 4 m long, with 2 m of grey space between corridors. The corridors were shown in a random order. The mice moved forward in the virtual reality corridors by running faster than a threshold of 6 cm s$^{-1}$, but the virtual corridors always moved at a constant speed (60 cm s$^{-1}$) as long as mice kept running faster than the threshold. Running was detected using an optical tracking sensor placed close to the ball.

The virtual reality corridors were created by concatenating four random crops from one of four large texture images: circle, leaf, rock and brick (Extended Data Fig. 1a). Grating stimuli (with angles of 0° and 45°) were also used to create the virtual reality corridors for a subset of mice ('unsupervised gratings' and VR$_g$ mice).

For behaviour-only experiments, each mouse was trained with rewards on one random pair of stimuli, such as leaf–circle or rock–brick. Mice with unsupervised pretraining (VR$_n$ mice) were pretrained on the same pair on which they were trained with rewards.

For imaging experiments, mice from both the task and the unsupervised cohorts were only trained on or exposed to one pair of stimuli, such as leaf–circle or rock–brick. For the mice exposed to the grating stimuli, we presented more than one pair of naturalistic stimuli before and after exposure to the gratings stimuli and recorded the neural responses to the naturalistic stimuli pairs, so each mouse can have more than one imaging session before and after. We also presented more than one pair of naturalistic stimuli to the naive mice, so each naive mouse can have more than one imaging session for testing naive responses.

The two different types of leaf1-swap stimuli (Extended Data Figs. 6f and 7a) were introduced separately for task mice but are pooled together for statistics analysis. The two leaf1-swap stimuli were introduced in the same session for unsupervised and naive mice, but were treated as two different data points and pooled together for statistical analyses.

**Water restriction procedure.** Water restriction procedures were conducted according to the IACUC. During the virtual reality + reward training condition, animals received an average of 1 ml water per day (range of 0.8–1.2 ml depending on health status and behavioural performance). Before reaching 1 ml water per day after the initiation of the restriction procedure, we gradually reduced the water amount from 2 ml per day to 1.5 ml per day until finally to 1 ml per day. The behaviour-only mice were water restricted for 5 days right before the virtual reality + reward training condition. Once the mice finished the virtual reality + reward training session, the remaining water (0.8–1.2 ml minus the amount received during experiment) was provided 0.5 h after the training. During the whole water restriction period, the body weight, appearance and behaviours were monitored using a standard quantitative health assessment system[58].

**Water reward delivery and lick detection.** A capacitance detector was connected with the metal lick port to detect licking. Mice received a drop of water (2.5 µl) if they correctly licked inside the reward corridor. In day 1 of the virtual reality + reward training session, we always delivered the water passively (passive mode) so that the mice could get used to acquiring reward when stimuli were present. For all the behaviour-only mice (Fig. 5) and some of the imaging mice (Figs. 1–4), we switched to active-reward mode after day 1 so that the mice had to lick within the reward zone to trigger the water delivery. For some of the imaging mice (Figs. 1–4), we kept using the passive mode but added a delay (1 s or 1.5 s) between the sound cue and reward delivery. Given that mice started licking as soon as they entered the corridor and until they received the water, adding a delay versus active-reward mode did not change how the mice behaved (Figs. 1–4).

**Behavioural training.** All animals were handled via refined handling techniques for at least 3 days before being acclimated to head fixation on the ball. Animals were acclimated gradually (0.5–1 h per day) on the ball over at least 3 days until they could be head fixed without exhibiting any signs of distress. Then, animals began a running training regiment (1 h per day), which lasted for at least 5 days to ensure they could run smoothly and continuously on the ball before being exposed to the

closed-loop virtual linear corridor. For water-restricted mice, we trained them for 2 days to get used to acquiring water from the spout when no stimulus was presented, before the virtual reality + reward training session. The $VR_n$ and $VR_g$ pretraining groups of mice (Fig. 5) were trained to acquire water from the spout on the last 2 days of unsupervised pretraining with no stimuli presented, after the virtual reality session ended to avoid the associative learning between stimuli and rewards. For the group without pretraining (Fig. 5), learning to get reward from the spout was similarly carried out after the running training session on the last 2 days of running training.

For the behaviour-only experiment, all animals started training in the virtual reality + reward training session on a Monday and continued training for exactly 5 days. The first day of training consisted of passive reward training, during which the reward was always delivered to the mouse in the rewarded corridor at the beginning of the reward zone. The next 4 days consisted of active reward training, during which the mice were required to lick in the reward zone to trigger the reward. This ensured a consistent training schedule during the critical learning period. The beginning of the reward zone was randomly chosen per trial from a uniform distribution between 2 m and 3 m, and continued until the end of the corridor. This meant that the earliest position in the rewarded corridor in which the mouse can receive the reward was a range of 2–3 m.

For imaging mice, the sound cue was presented in all trial types, for task and unsupervised mice, and the time of the sound cue was randomly chosen per trial from a uniform distribution between positions 0.5 m and 3.5 m. For task mice, the sound cue indicated the beginning of the reward zone in the rewarded corridor. The reward was delivered if a lick was detected after the sound cue in the rewarded corridor. As mice kept licking (anticipatory licking) as soon as they entered the reward corridors and learned to lick right after the sound cue, the reward delivery locations were also approximately uniformly distributed between 0.5 m and 3.5 m (Fig. 1c). In some mice, the reward was delivered passively with a delay after the sound cue, but these mice still showed anticipatory licking before the sound cue. To rule out whether the licking was due to detecting signals related to the reward delivery, such as water coming out the spout or sound produced by solenoid valve, we considered a lick response if the mouse licked at least once inside the corridor but before the sound cue. Although the rewards were absent, the sound cue was still presented in the unsupervised training experiment for consistency.

## Data analysis

For analysis, we used Python 3 (ref. 59), primarily based on numpy and scikit-learn[60,61], as well as Rastermap[39]. The figures were made using matplotlib and jupyter-notebook[62,63].

**Processing of calcium imaging data.** Calcium imaging data were processed using Suite2p[32], available on GitHub (www.github.com/MouseLand/suite2p). Suite2p performs motion correction, region of interest detection, cell classification, neuropil correction and spike deconvolution as previously described[64]. For non-negative deconvolution, we used a timescale of decay of 0.75 s (ref. 65). All our analyses were based on deconvolved fluorescence traces.

**Neural selectivity ($d'$).** To compute the selectivity index $d'$, illustrated in Fig. 1f, we only selected data points inside the 0–4-m region of the corridors where the textures were shown. We excluded the data points in which the animal was not running, so that all data points included for calculating the selectivity index came from similar engagement or arousal levels of the mice. Note that these data points are computed from original estimated deconvolved traces without interpolation. We first calculated the means ($\mu_1$ and $\mu_2$) and standard deviations ($\sigma_1$ and $\sigma_2$) of activities for any two corridors, then computed the $d'$. The criteria for selective neurons was $|d'| \geq 0.3$:

$$d' = \frac{\mu_1 - \mu_2}{\frac{\sigma_1}{2} + \frac{\sigma_2}{2}}$$

To make the density plots across the cortex (for example, Fig. 1i), we computed 2D histograms for each session based on the selective neurons in that session. We then applied a 2D Gaussian filter to this matrix and divided by the number of total recorded neurons in that session to get a density map for each mouse. Before averaging the density maps across mice, we assigned NaN to areas where no neurons were recorded. This ensured no underestimation on the density within areas where not all mice have neurons recorded.

For sequence similarity analyses (Fig. 2 and Extended Data Figs. 3 and 6), we used half of leaf1 and circle1 trials (train trials) to compute the selectivity index $d'$ and we selected neurons based on the criteria $|d'| \geq 0.3$. We then split the other half of leaf1 and circle1 trials (test trials) into odd versus even trials to compute spatial tuning curves for odd and even trials separately for each selective neuron. From these spatial tuning curves, we used the position with the maximal response as the preferred position for each neuron. To compute tuning curves for other stimuli such as leaf2, circle2 and swap (which were not used to find selective neurons), we split all trials into odd and even trials. The preferred positions of the same neurons in different corridors or in odd versus even trials were used to compute a correlation coefficient ($r$).

**Coding direction and similarity index.** To compute the coding direction (Figs. 2 and Fig. 3 and Extended Data Figs. 3, 6 and 7), for example, in leaf1 versus circle1, we first chose leaf1-selective and circle1-selective neurons based on their $d'$ from the train trials (using the top 5% selective neurons each to leaf1 and circle1, same as the sequence similarity analysis). Then, we acquired the neural activity for every position through interpolation, and normalized the neural activity **r** for each neuron by subtracting the baseline response in the grey portion of the corridor $\mu_{grey}$, and dividing by the average standard deviation of the responses of a neuron in each corridor:

$$\mathbf{r}_{norm} = \frac{\mathbf{r} - \mu_{grey}}{\frac{\sigma_{leaf1}}{2} + \frac{\sigma_{circle1}}{2}}$$

We then computed the mean normalized activity $\mu_{leaf1}$ of leaf1-selective neurons and the mean normalized activity $\mu_{circle1}$ of circle1-selective neurons at each position in each corridor. The coding direction $\mathbf{v}_t^{proj}$ on a given trial $t$ was defined as the difference

$$\mathbf{v}_t^{proj} = \mu_{leaf1} - \mu_{circle1}$$

Note that this is equivalent to assigning weights of $\frac{1}{N\mathrm{trials}_{leaf1}}$, $\frac{-1}{N\mathrm{trials}_{circle1}}$ and 0, respectively for positively selective, negatively selective and non-selective neurons, and using those weights as a projection vector for the neural data. We investigated the coding direction always on test trials not used for selecting neurons, either from held-out trials of leaf1 and circle1, or for trials with other stimuli. We averaged the responses across each trial type: $\mathbf{v}_{leaf1}^{proj} = \sum_{t \in leaf1}^{N_{leaf1}} \mathbf{v}_t^{proj}$ (for example, Fig. 2i, left).

Average projections for each trial type were computed by averaging these projections within the texture area (0–4 m) (for example, Fig. 2i, right), denoted as $a_{leaf1}^{proj}$, $a_{leaf2}^{proj}$, $a_{circle1}^{proj}$ and $a_{circle2}^{proj}$. We then defined the similarity index (SI) on a per-stimulus basis, for example, for leaf2, as:

$$dy = a_{leaf1}^{proj} - a_{leaf2}^{proj}$$
$$dx = a_{leaf2}^{proj} - a_{circle1}^{proj}$$
$$SI_{leaf2} = \frac{dx - dy}{dx + dy} \qquad (-1 \leq SI \leq 1)$$

which is quantified in Fig. 2j. We also computed the coding direction for different sets of selective neurons, for example, leaf1 versus leaf2, and

then computed the similarity indices for leaf3 and circle1 (Extended Data Fig. 6c,d).

**Reward-prediction neurons.** Reward-prediction neurons were either selected using the clustering algorithm Rastermap (Fig. 4a–d) or using a $d'$ criterion (Fig. 4e–n). Using Rastermap, we selected the reward-prediction neurons based on their special firing patterns of only responding inside the rewarded corridor and specifically before reward delivery. Using $d'$, we first interpolated the neural activity of single neurons based on their position inside the corridor and constructed a matrix (trials by positions). Only the leaf1 trials (rewarded for task mouse cohort, and unrewarded for the unsupervised cohort) were chosen and divided into early-cue trials versus late-cue trials based on the sound cue position inside the corridor. We used cue position instead of reward position because the sound cues were played in each corridor at a random position, with or without reward, and these sound cue positions were highly correlated with reward positions in the rewarded corridor (Fig. 1c). We then calculated the $d'_{\text{late vs early}}$ as:

$$d'_{\text{late vs early}} = \frac{\mu_{\text{late}} - \mu_{\text{early}}}{\frac{\sigma_{\text{late}}}{2} + \frac{\sigma_{\text{early}}}{2}}$$

and selected the reward-prediction neurons with $d'_{\text{late vs early}} \geq 0.3$. The activity of the reward-prediction neural population in Fig. 4 (except Fig. 4d) was acquired following $k$-fold cross-validation. We randomly split all trials into tenfolds. we used ninefolds as training trials to compute $d'_{\text{late vs early}}$. Trial-by-trial activity for the remaining onefold (test trials) was computed by averaging across the reward-prediction neurons that met the selection criteria. We repeated this ten times until average population activity for every fold (and thus every trial) was acquired.

To obtain reward-prediction activity aligned to the first lick (Fig. 4j), only rewarded trials (leaf1) with a first lick happening after 2 m from corridor entry were included to enable us to investigate the reward-prediction signal before licking starts. Owing to this criteria, one mouse was excluded because there were no trials with a first lick later than 2 m.

To obtain reward-prediction activity and activity of leaf1-selective neurons in leaf2 trials (Fig. 4k,l), one mouse was excluded due to having only one leaf2 trial without licking.

**Running speed.** To compare the running speed before and after learning, we selected the period when mice were running faster than 6 cm s$^{-1}$ for at least 66 ms (a threshold that triggers the motion of the virtual reality; Extended Data Fig. 2d). For Extended Data Fig. 8e, the running speed was interpolated to the timepoints of the imaging frames using the function scipy.interpolate.interp1d. For Extended Data Fig. 2a–c, the running speed for every position (0–6 m, with a 0.1-m step size) was also acquired through the same interpolation method. Extended Data Fig. 2c shows the averaged running speed inside the texture areas (0–4 m).

**Statistics and reproducibility.** We performed paired, two-sided Student's $t$-tests in Figs. 1d,j, 2f,j, 3b,c, 4g,k,l and Extended Data Figs. 1f,g, 2c,d, 4a,c,d, 5b, 6d,i, 7d,g, 8a,c and 9; and performed independent two-sided Student's $t$-tests in Figs. 3e,h and 5f. Statistical significance was calculated as *$P < 0.05$, **$P < 0.01$ and ***$P < 0.001$. No adjustments were made for multiple comparisons. Error bars on all figures represent s.e.m. The exact $P$ values are below for each figure. Where four values are reported, these are for V1, medial, lateral and anterior regions.

- Fig. 1d: 0.714 before learning and $5.97 \times 10^{-4}$ after learning
- Fig. 1j: 0.940, 0.00726, 0.0341 and 0.0261 for task mice; 0.212, $3.21 \times 10^{-4}$, 0.245 and 0.0202 for unsupervised mice; and 0.146, 0.318, 0.0632 and 0.655 for unsupervised grating mice

- Fig. 2f: $7.97 \times 10^{-6}$, $2.56 \times 10^{-4}$, $1.84 \times 10^{-5}$ and $4.1 \times 10^{-3}$ for task mice; $1.72 \times 10^{-7}$, $1.04 \times 10^{-4}$, $1.25 \times 10^{-4}$ and $3.07 \times 10^{-3}$ for unsupervised mice; and $2.11 \times 10^{-10}$, $1.25 \times 10^{-5}$, $2.55 \times 10^{-8}$ and $8.661 \times 10^{-6}$ for naive mice
- Fig. 2j: 0.0015, $2.94 \times 10^{-4}$, 0.0020 and 0.0013 for task mice; $1.72 \times 10^{-4}$, $7.24 \times 10^{-5}$, $2.44 \times 10^{-4}$ and 0.0082 for unsupervised mice; and $2.47 \times 10^{-6}$, $5.97 \times 10^{-9}$, $2.35 \times 10^{-7}$ and $2.73 \times 10^{-6}$ for naive mice
- Fig. 3b: 0.0037, 0.0922, 0.0026 and 0.0136 task mice; and $2.34 \times 10^{-5}$, 0.0081, $2.96 \times 10^{-4}$ and 0.0146 for unsupervised mice
- Fig. 3c: 0.0074
- Fig. 3e: $P_{\text{supervised vs naive}}$: 0.991, $1.381 \times 10^{-5}$, 0.352 and 0.053; $P_{\text{unsupervised vs naive}}$: 0.797, $4.45 \times 10^{-6}$, 0.282 and 0.727; and $P_{\text{unsupervised grating vs naive}}$: 0.226, 0.284, 0.239 and 0.570
- Fig. 3h: $P_{\text{supervised vs naive}}$: 0.084, 0.002, $2.81 \times 10^{-4}$ and $6.2 \times 10^{-7}$; $P_{\text{unsupervised vs naive}}$: $5.26 \times 10^{-5}$, $1.18 \times 10^{-5}$, $9.32 \times 10^{-5}$ and 0.011; and $P_{\text{unsupervised grating vs naive}}$: 0.316, 0.705, 0.375 and 0.945
- Fig. 4g: 0.0069 for task mice and 0.708 for unsupervised mice
- Fig. 4k: 0.014
- Fig. 4l: 0.180
- Fig. 5f. $P_{VR_n \text{ vs no pretraining}}$ 1 by day: 0.00493, 0.00555, 0.0554, 0.259 and 0.579; $P_{VR_n \text{ vs } VR_g}$ by day: 0.538, 0.0348, 0.221, 0.541 and 0.978; and $P_{VR_g \text{ vs no pretraining}}$ by day: 0.00782, 0.148, 0.415, 0.476 and 0.510
- Extended Data Fig. 1f (left): 0.206, 0.00497, 0.0297 and 0.0114 for task mice; and 0.0389, $1.15 \times 10^{-4}$, 0.982 and 0.249 for unsupervised mice
- Extended Data Fig. 1f (right): 0.562, 0.0219, 0.104 and 0.134 for task mice; and 0.017, 0.00130, 0.0129 and 0.00977 for unsupervised mice
- Extended Data Fig. 1g (left): 0.419 for task mice and 0.0466 for unsupervised mice
- Extended Data Fig. 1g (right): 0.574 for task mice and 0.126 for unsupervised mice
- Extended Data Fig. 2c (left): 0.883 for circle1 and 0.660 for leaf1
- Extended Data Fig. 2c (right): 0.154 for circle1 and 0.724 for leaf1
- Extended Data Fig. 2d (left): 0.814 for circle1 and 0.533 for leaf1
- Extended Data Fig. 2d (right): 0.017 for circle1 and 0.923 for leaf1
- Extended Data Fig. 4a: $P_{\text{circle1}} = 0.163$, $P_{\text{circle2}} = 0.923$, $P_{\text{leaf1}} = 0.028$ and $P_{\text{leaf2}} = 0.013$
- Extended Data Fig. 4c (V1): $P_{\text{circle1}} = 0.492$, $P_{\text{circle2}} = 0.106$, $P_{\text{leaf1}} = 0.018$ and $P_{\text{leaf2}} = 0.041$
- Extended Data Fig. 4c (medial): $P_{\text{circle1}} = 0.947$, $P_{\text{circle2}} = 0.190$, $P_{\text{leaf1}} = 0.474$ and $P_{\text{leaf2}} = 0.072$
- Extended Data Fig. 4c (lateral): $P_{\text{circle1}} = 0.936$, $P_{\text{circle2}} = 0.150$, $P_{\text{leaf1}} = 0.046$ and $P_{\text{leaf2}} = 0.326$
- Extended Data Fig. 4c (anterior): $P_{\text{circle1}} = 0.316$, $P_{\text{circle2}} = 0.112$, $P_{\text{leaf1}} = 0.013$ and $P_{\text{leaf2}} = 0.935$
- Extended Data Fig. 4d: $P_{V1} = 0.531$, $P_{\text{medial}} = 0.808$, $P_{\text{lateral}} = 0.308$ and $P_{\text{anterior}} = 0.015$
- Extended Data Fig. 5b: 0.772, 0.799, 0.474 and 0.978 for task mice; and 0.957, 0.476, 0.904 and 0.691 for unsupervised mice
- Extended Data Fig. 6d: 0.0034, $5.46 \times 10^{-4}$, 0.0059 and 0.082 for task mice; 0.019, 0.011, 0.0057 and 0.094 for unsupervised mice; 0.486, 0.654, 0.130 and 0.352 for naive mice; and 0.414, 0.217, 0.077 and 0.307 for unsupervised grating mice
- Extended Data Fig. 6i: $6.18 \times 10^{-4}$, $8.79 \times 10^{-4}$, 0.0015 and 0.082 for task mice; $3.23 \times 10^{-6}$, $5.34 \times 10^{-6}$, $1.44 \times 10^{-6}$ and $7.51 \times 10^{-6}$ for unsupervised mice; and $6.45 \times 10^{-7}$, $1.99 \times 10^{-4}$, $5.14 \times 10^{-5}$ and 0.0040 for naive mice
- Extended Data Fig. 7d: $2.06 \times 10^{-6}$, $5.47 \times 10^{-4}$, $2.31 \times 10^{-4}$ and $1.41 \times 10^{-4}$ for task mice; $4.75 \times 10^{-9}$, $3.30 \times 10^{-5}$, $1.25 \times 10^{-7}$ and $2.62 \times 10^{-5}$ for unsupervised mice; and $4.71 \times 10^{-10}$, 0.881, $3.94 \times 10^{-6}$ and $3.33 \times 10^{-7}$ for naive mice
- Extended Data Fig. 7g: $P_{\text{leaf1 vs leaf2}} = 0.347$
- Extended Data Fig. 8a: 0.496, 0.151, 0.091 and 0.0069 for task mice; and 0.441, 0.632, 0.882 and 0.708 for unsupervised mice
- Extended Data Fig. 8c: 0.277, 0.700, 0.548 and 0.0210 for task mice; and 0.183, 0.276, 0.235 and 0.0546 for unsupervised mice
- Extended Data Fig. 9 (day 1): 0.0166 for $VR_n$ pretraining, 0.157 for $VR_g$ pretraining and 0.272 for no pretraining

- Extended Data Fig. 9 (day 2): 0.0033 for $VR_n$ pretraining, 0.0088 for $VR_g$ pretraining and 0.0823 for no pretraining
- Extended Data Fig. 9 (day 3): 0.0120 for $VR_n$ pretraining, 0.188 for $VR_g$ pretraining and 0.0050 for no pretraining
- Extended Data Fig. 9 (day 4): 0.067 for $VR_n$ pretraining, 0.0356 for $VR_g$ pretraining and 0.0180 for no pretraining
- Extended Data Fig. 9 (day 5): 0.1185 for $VR_n$ pretraining, 0.0230 for $VR_g$ pretraining and 0.0317 for no pretraining.

**Retinotopy.** Retinotopic maps for each imaging mouse were computed based on receptive field estimation using neural responses to natural images (at least 500 natural images repeated 3 times each). This proceeded in several steps:

(1) Obtained a well-fit convolutional encoding model of neural responses with an optimized set of 200 spatial kernels, using a reference mouse (Extended Data Fig. 1b).
(2) Fitted all neurons from our imaging mice to these kernels to identify the preferred kernel and the preferred spatial position (Extended Data Fig. 1c).
(3) Aligned the spatial position maps to a single map from the reference mouse.
(4) Outlined the brain regions in the reference mouse using spatial maps and approximately following the retinotopic maps from ref. 33.

Compared with previous approaches, ours took advantage of single-neuron responses rather than averaging over entire local populations, and by using natural images that we can better drive neurons and obtain their specific receptive field models. The mapping procedure was sufficiently efficient that it could be performed in a new mouse with responses to only 500 test images each repeated 3 times. Below we describe each step in detail:

For step 1, using the reference mouse, we used the following to model the response of neuron $n$ to image img:

$$F_n(\text{img}) = a_n \cdot (K \circ \text{img})(k_n, x_n, y_n)$$

where $a_n$ is a positive scalar amplitude, $\circ$ represents the convolution operation, $x_n$ and $y_n$ represent the position in the convolution map for neuron $n$, $k_n$ represents the index of the convolutional map, and $K$ is a matrix of size $200 \times 13 \times 13$ containing the convolutional filters. This model was fit to neural responses to a natural image dataset of approximately 5,000 images shown at a resolution of $120 \times 480$, which were downsampled to $30 \times 120$ for fitting. The kernels $K$ were initialized with random Gaussian noise. An iterative expectation maximization-like algorithm was used to optimize the kernels, which alternated between: (1) finding the best position $(x_n, y_n)$ for each neuron $n$, as well as the best kernel $k_n$ and the best amplitude $a_n$; and (2) optimizing $K$ given a fixed assignment $(x_n, y_n, k_n, a_n)$ for each neuron $n$. The first part of the iteration was done in a brute force manner: responses of each kernel at each location for each image were obtained and correlated with the responses of each neuron. The highest correlated match for each neuron was then found and its corresponding $(x_n, y_n, k_n, a_n)$ was used to fit $K$. The best estimate for kernels $K$ was approximately equivalent to averaging the linear receptive field all cells $n$ assigned to a kernel $k_n$ after alignment to their individual spatial centres $x_n, y_n$. After each iteration, the kernels were translated so their centres of mass would be centred in the $13 \times 13$ pixel frame. The centre of mass was obtained after taking the absolute value of the kernel coefficients. After less than ten iterations, the kernels converged to a set of well-defined filters (Extended Data Fig. 1b).

For step 2, after the kernels $K$ were estimated once, for a single-reference recording, we used them for all recordings by repeating the first step of the iterative algorithm in step 1 with a slight modification. Instead of assigning each neuron $(x_n, y_n, k_n, a_n)$ independently,

we averaged the 2D, maximum correlation maps of the nearest 50 neurons to each neuron, and then took their maximum. This essentially smoothed the spatial correlations to ensure robust estimation even for neurons with relatively little signal (Extended Data Fig. 1c).

For step 3, to align spatial maps to the reference mouse, we used kriging interpolation to find a tissue-to-retinotopy transformation $f$. Intuitively, we wanted to model the data from the alignment mouse as a smooth function $f$ from a 2D space of $(z, t)$ positions in tissue to another 2D space of retinotopic preferences $(x, y)$. For a new mouse with tissue positions $(z', t')$ and retinotopic positions $(x', y')$, we could then optimize an affine transform $A$ composed of a $2 \times 2$ matrix $A_1$ and $1 \times 2$ bias term $A_2$ such that

$$(z'_a, t'_a) = A_1 \cdot (z', t') + A_2$$

so that

$$\text{Cost} = \|f(z'_a, t'_a) - (x', y')\|^2$$

is minimized. To fit the smooth function $f$, we used kriging interpolation, so that $f$ is the kriging transform

$$f(z'_a, t'_a) = F((z'_a, t'_a), (z, t)) \cdot F((z, t), (z, t))$$
$$\cdot \text{Cov}((z, t), (x, y)),$$

where $F$ is a squared exponential kernel $F(a, b) = \exp(-\|a - b\|^2 / \sigma^2)$ with a spatial constant $\sigma$ of 200 μm and Cov is the covariance between inputs and outputs. Note that we could precompute the second part of $f$ as it does not depend on $(zz'_a, t'_a)$. We then optimized the affine transform $A$. $A$ was initialized based on a grid search over possible translation values within ±500 μm. After the grid search, we used gradient descent on the values of $A$, allowing for translation and rotation, but with a regularization term on $A_1$ to keep the matrix close to the identity. Finally, for some sessions, the optimization did not converge, in which case we restricted the matrix $A_1$ to a fixed determinant, thus preventing a scaling transform.

For step 4, this final step was to delineate area borders on the reference mouse, which were then transformed to all mice as described in step 3. Similar to ref. 33, we computed the sign map and parcellated it into regions where the sign did not change. Ambiguities in the sign map were resolved by approximately matching areas to the data from ref. 33. Note that the exact outlines of the areas in some cases had different shapes from those in ref. 33. This is to be expected from two sources: (1) the maps in ref. 33 were computed from widefield imaging data, which effectively blurs over large portions of the cortex, thus obscuring some boundaries and regions; and (2) our specific cranial windows are in a different position from ref. 33. Nonetheless, we do not think the small mismatch in area shapes would have a large effect on our conclusions, given that we combined multiple areas into large regions.

## Reporting summary

Further information on research design is available in the Nature Portfolio Reporting Summary linked to this article.

## Data availability

The data are available on Janelia (https://doi.org/10.25378/janelia.28811129.v1)[66].

## Code availability

The code is available on GitHub (https://github.com/MouseLand/zhong-et-al-2025)[67].

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

**Acknowledgements** This research was funded by the Howard Hughes Medical Institute at the Janelia Research Campus. We thank J. Cox, C. Lopez, A. Kuzspit, M. Rose, M. Michaelos, G. Harris, S. Lindo and their respective teams from Vivarium for animal breeding, husbandry, surgeries and behavioural training support; A. Sohn, T. Goulet, D. Tsyboulski and S. Sawtelle from JeT for help with rig maintenance and upgrades; G. Jaindl, M. Sandoe and B. Djiguemde from MBF Bioscience for ScanImage support; and S. Romani, V. Jayaraman and N. Spruston for helpful discussions about the work.

**Author contributions** L.Z., C.S. and M.P. designed the study. L.Z. performed the imaging experiments. L.Z., S.B. and R.G. performed the behavioural experiments. L.Z. and J.A. designed and maintained the behavioural apparatus. D.F. maintained and improved the microscope. L.Z., C.S. and M.P. performed the data analysis. L.Z., C.S. and M.P. wrote the manuscript with input from all authors.

**Competing interests** The authors declare no competing interests.

**Additional information**
**Correspondence and requests for materials** should be addressed to Lin Zhong or Marius Pachitariu.

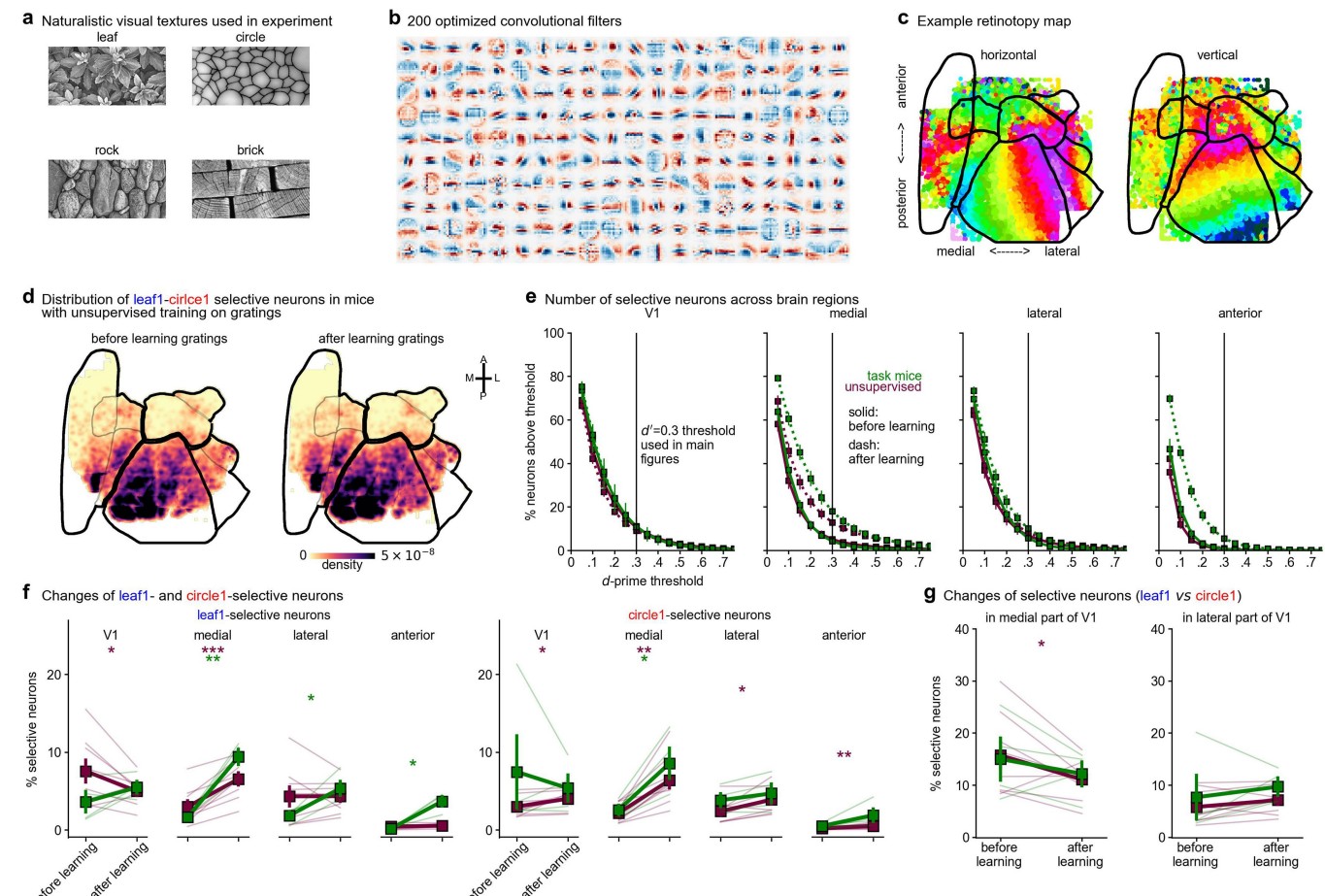

**a** Naturalistic visual textures used in experiment
leaf · circle · rock · brick

**b** 200 optimized convolutional filters

**c** Example retinotopy map
horizontal · vertical
anterior / posterior · medial / lateral

**d** Distribution of leaf1-cirlce1 selective neurons in mice with unsupervised training on gratings
before learning gratings · after learning gratings
0 — density — 5 × 10⁻⁸

**e** Number of selective neurons across brain regions
V1 · medial · lateral · anterior
% neurons above threshold
$d'$=0.3 threshold used in main figures
task mice / unsupervised
solid: before learning
dash: after learning
$d$-prime threshold

**f** Changes of leaf1- and circle1-selective neurons
leaf1-selective neurons: V1 * / medial *** ** / lateral * / anterior *
circle1-selective neurons: V1 * / medial ** * / lateral * / anterior **
% selective neurons · before learning · after learning

**g** Changes of selective neurons (leaf1 vs circle1)
in medial part of V1 * · in lateral part of V1
% selective neurons · before learning · after learning

**Extended Data Fig. 1 | Retinotopy and neural changes after learning for different populations. a**, Example stimulus crops. **b**, Convolutional filter bank used for mapping retinotopy. **c**, Retinotopic maps for an example mouse after alignment to a reference atlas. **d**, The distribution of selective neurons for leaf1 and circle1 for mice with unsupervised exposure to grating stimuli. Similar to Fig. 1i. **e**, Number of selective neurons as a function of threshold. **f**, Same as Fig. 1j split into leaf1- selective (left) and circle1- selective neurons (right). **g**, Similar to Fig. 1j for V1 neurons split into a medial and a lateral part. All data are mean ± s.e.m. *$P$<0.05, **$P$<0.01, ***$P$<0.001.

**a** Example unsupervisd mouse's running speed

**b** Example task mouse's running speed

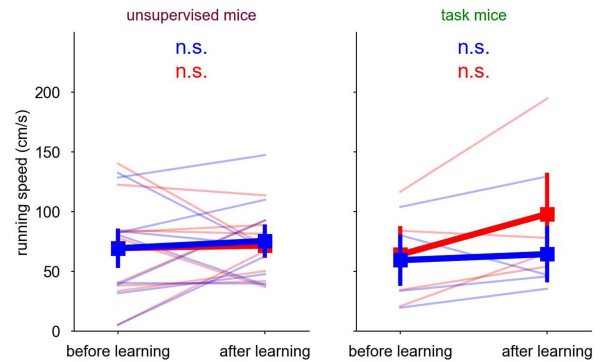

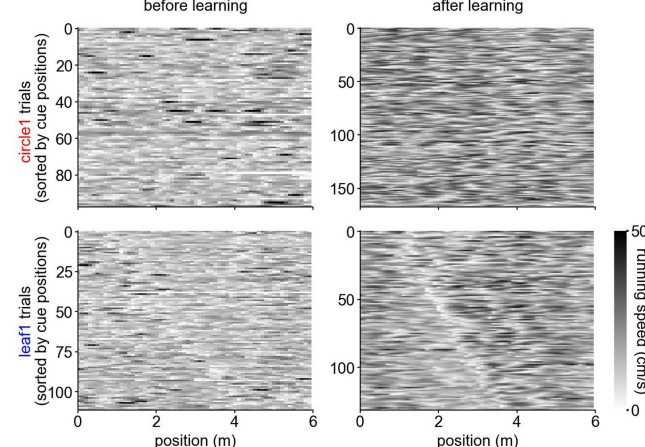

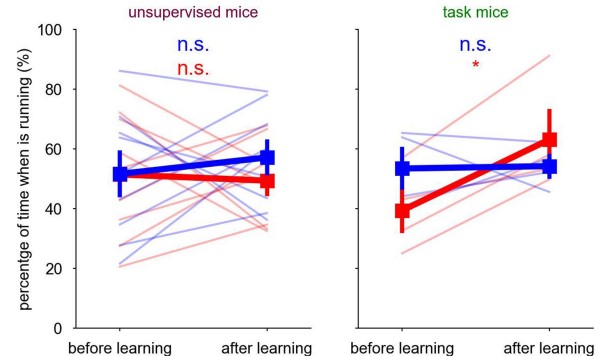

**c** Summary of running speed in circle1 and leaf1

**d** Summary of running state

**Extended Data Fig. 2 | Running behaviors. a**, Running speed of example mouse from unsupervised experiment with imaging. **b**, Same as **a**, for example task mouse. **c**, Summary of running speeds before vs. after learning (task mice:

n=4 mice; unsupervised: n=9 mice). **d**, Percentage of time when mice were running. All data are mean ± s.e.m. *P<0.05, n.s., not significant.

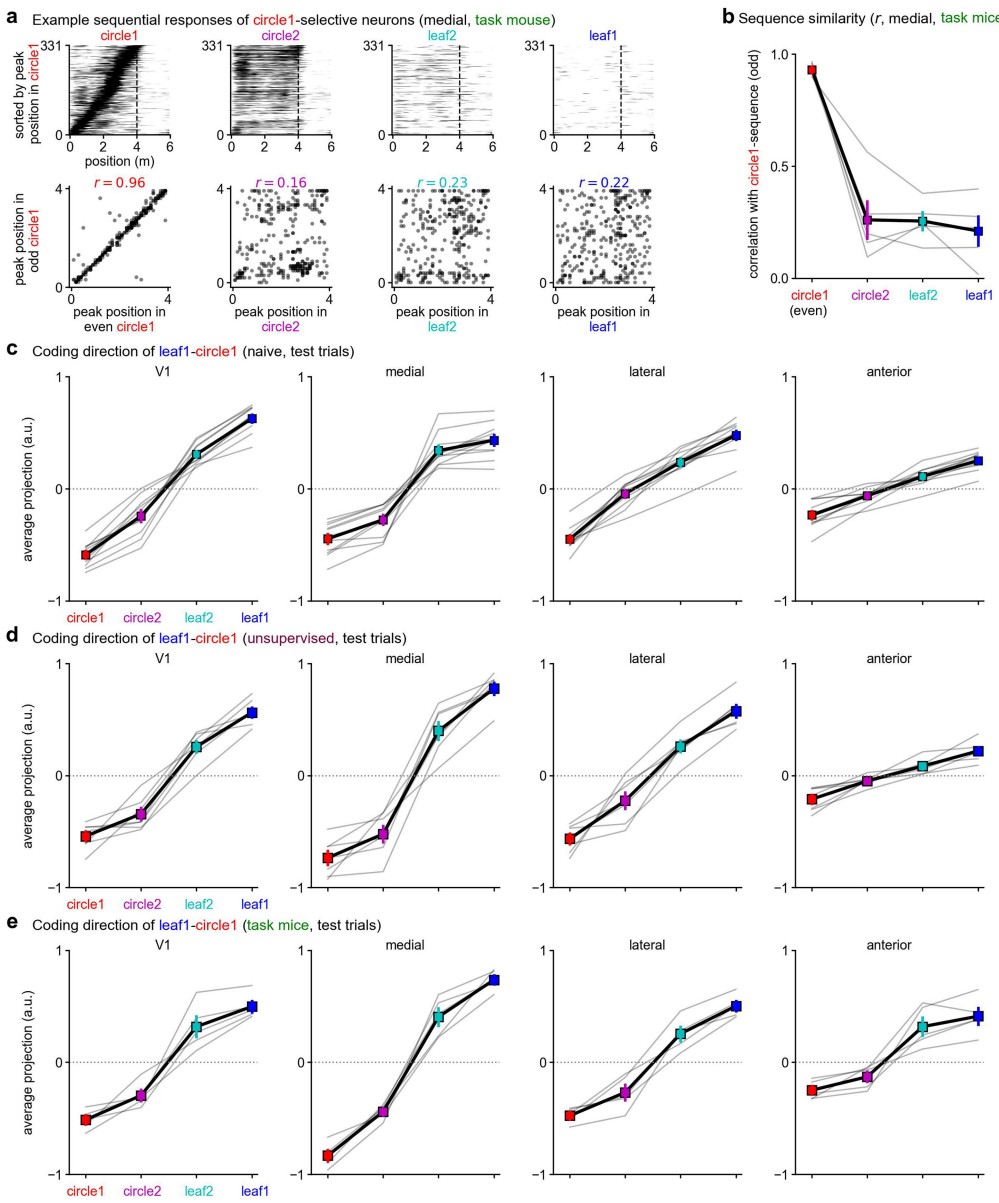

**Extended Data Fig. 3 | Sequences in circle1-preferring neurons and average projections on the coding direction. a**, Same as Fig. 2d for circle1-selective neurons sorted by the circle1 trials. **b**, Same as Fig. 2e for circle1-selective neurons. **c-e** Same as Fig. 2i for all brain regions and all mouse cohorts: **c**, naive (n=9 mice, 11 sessions), **d**, unsupervised (n=7 mice), **e**, task mice (n=5 mice).

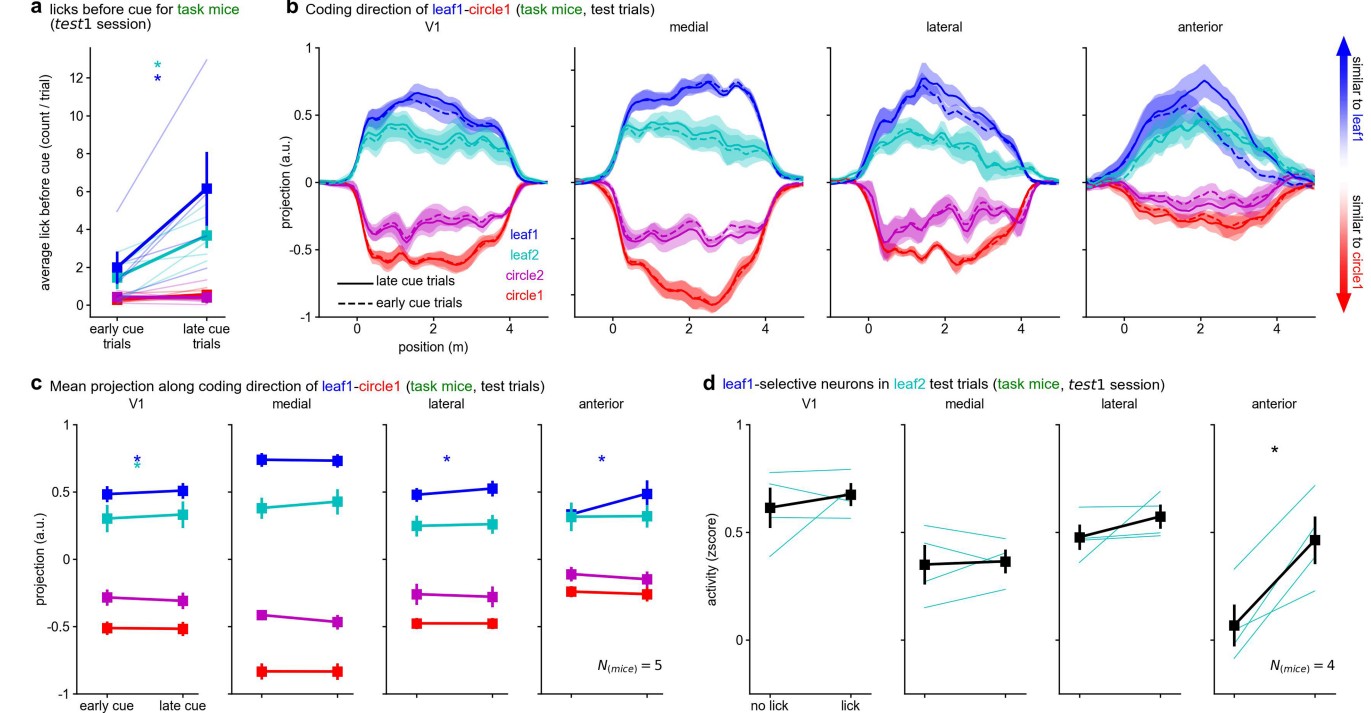

**Extended Data Fig. 4 | Relation between neural activity and licking behaviors. a**, Average licks per trial before sound cue in early and late cue trials (*test1* session, task mice: n=5). **b**, Coding direction projections for early and late cue trials (*test1*). **c**, Same as **b** but averaged over 0-4 meters. **d**, Same as Fig. 4l for all regions. Lick trials contain at least one lick inside the entire corridor. All data are mean ± s.e.m. *P<0.05.

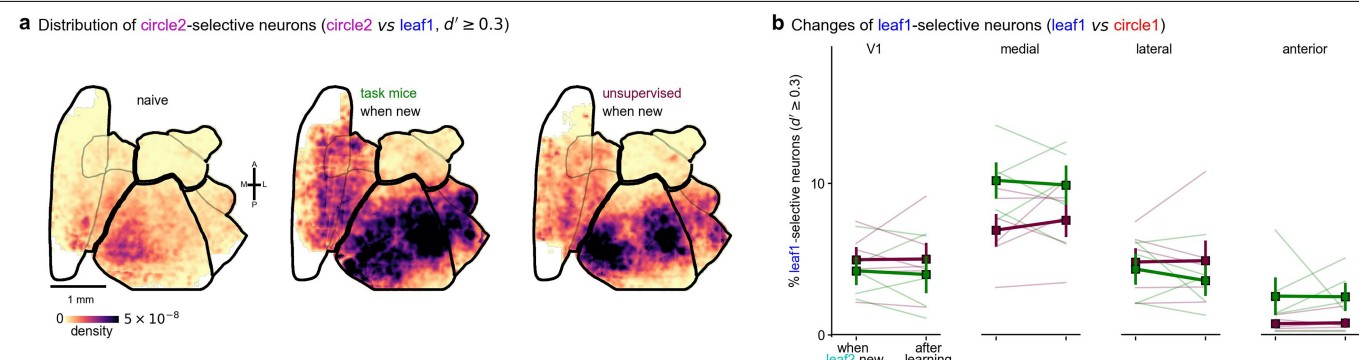

**a** Distribution of circle2-selective neurons (circle2 *vs* leaf1, $d' \geq 0.3$)

naive

task mice when new

unsupervised when new

1 mm

0 $5 \times 10^{-8}$
density

**b** Changes of leaf1-selective neurons (leaf1 *vs* circle1)

V1 medial lateral anterior

% leaf1-selective neurons ($d' \geq 0.3$)

when leaf2 new after learning leaf2

**Extended Data Fig. 5 | Representations of familiar versus novel stimuli. a**, Same as Fig. 3a for circle2-selective neurons. **b**, Same as Fig. 3b for leaf1-selective neurons.

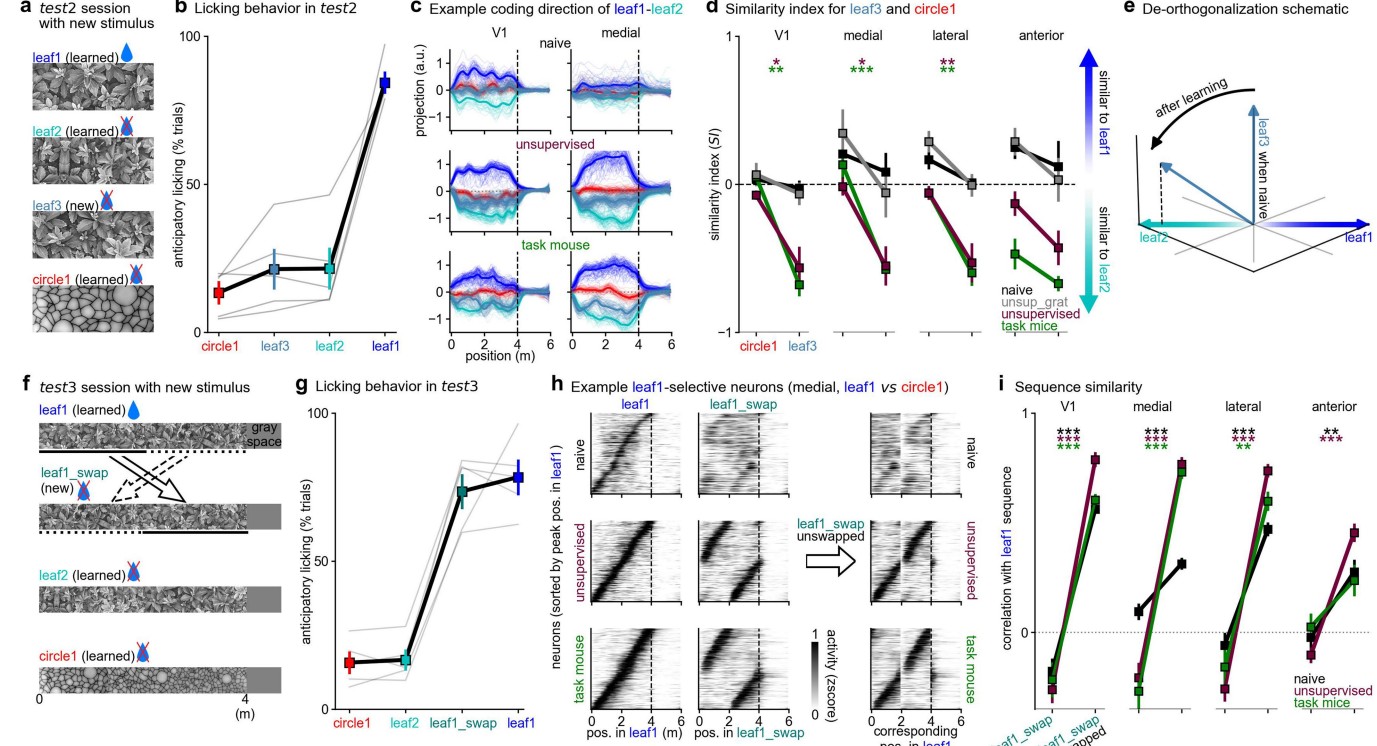

**Extended Data Fig. 6 | Visual recognition memory becomes exemplar-specific after extended training. a**, Stimuli in the *test2* session (see timeline in Fig. 1b). **b**, Anticipatory licking behavior in *test2* (n=5 mice). **c**, Example projections of neural data onto the coding direction of leaf1 vs leaf2. **d**, Similarity index (from Fig. 2i) of leaf3 and circle1 stimuli for the leaf1 vs leaf2 coding direction (task mice: n=5 mice; unsupervised: n=6 mice; naive: n=7 mice; unsupervised grating: n=3 mice, 5 sessions). **e**, Schematic of observed 'de-orthogonalization' effect, where an initially symmetric projection of leaf3 becomes asymmetric and more similar to the leaf2 neural vector. Neural vectors are referenced with respect to the center of the leaf1-leaf2 axis. **f**, Stimuli in the *test3* session

(see timeline in Fig. 1b). **g**, Anticipatory licking behavior in *test3* (n= 5 sessions in 3 mice). **h**, Example leaf1-selective neurons (medial region) during leaf1 and swap trials, sorted by responses in the leaf1 corridor for naive, task and unsupervised mice. Also shown (right) are the responses on swap trials after reversing the swap manually ('unswapped'). **i**, Average correlation in position preference between leaf1 and the swap as well as the unswapped responses, shown for all three groups of mice across regions (task mice: n=3 mice, 5 sessions; unsupervised: n=4 mice, 8 sessions; naive: n=3 mice, 10 sessions). All data are mean ± s.e.m. *P<0.05, **P<0.01, ***P<0.001.

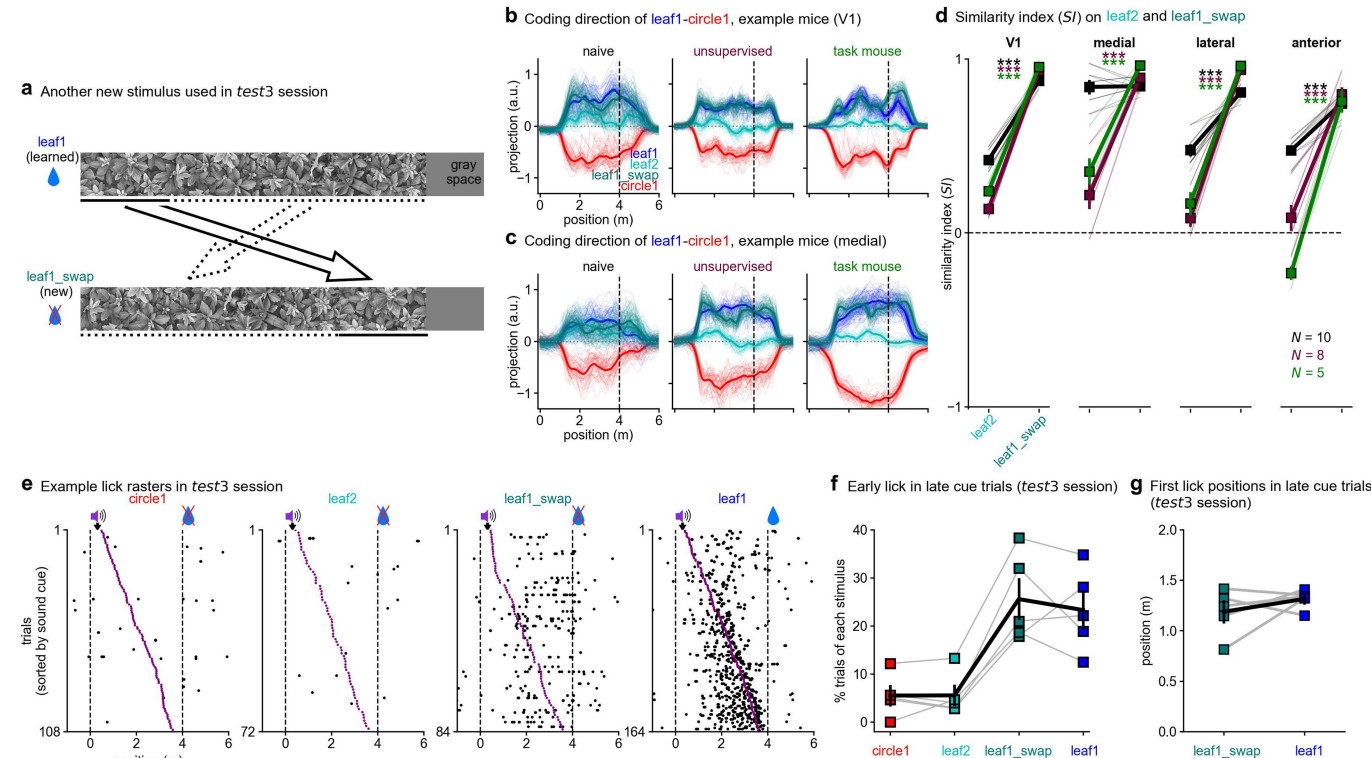

**Extended Data Fig. 7 | Coding direction projections and similarity indices during *test3*. a**, A second swap type used in the *test3* session. **b**, Example coding direction of leaf1 vs circle1 in the *test3* session in V1. **c**, Same as **b** in the medial region. **d**, Similarity index along the leaf1-circle1 coding direction in the *test3* session (task mice: n=3 mice, 5 sessions; unsupervised: n=4 mice, 8 sessions;

naive: n=3 mice, 10 sessions). **e**, Lick rasters for an example mouse (*test3*). **f**, Fraction of trials with early licks in late cue trials (>2m cue) (*test3 session*). **g**, Average position of first lick in late cue trials (*test3*). All data are mean ± s.e.m. ***$P < 0.001$.

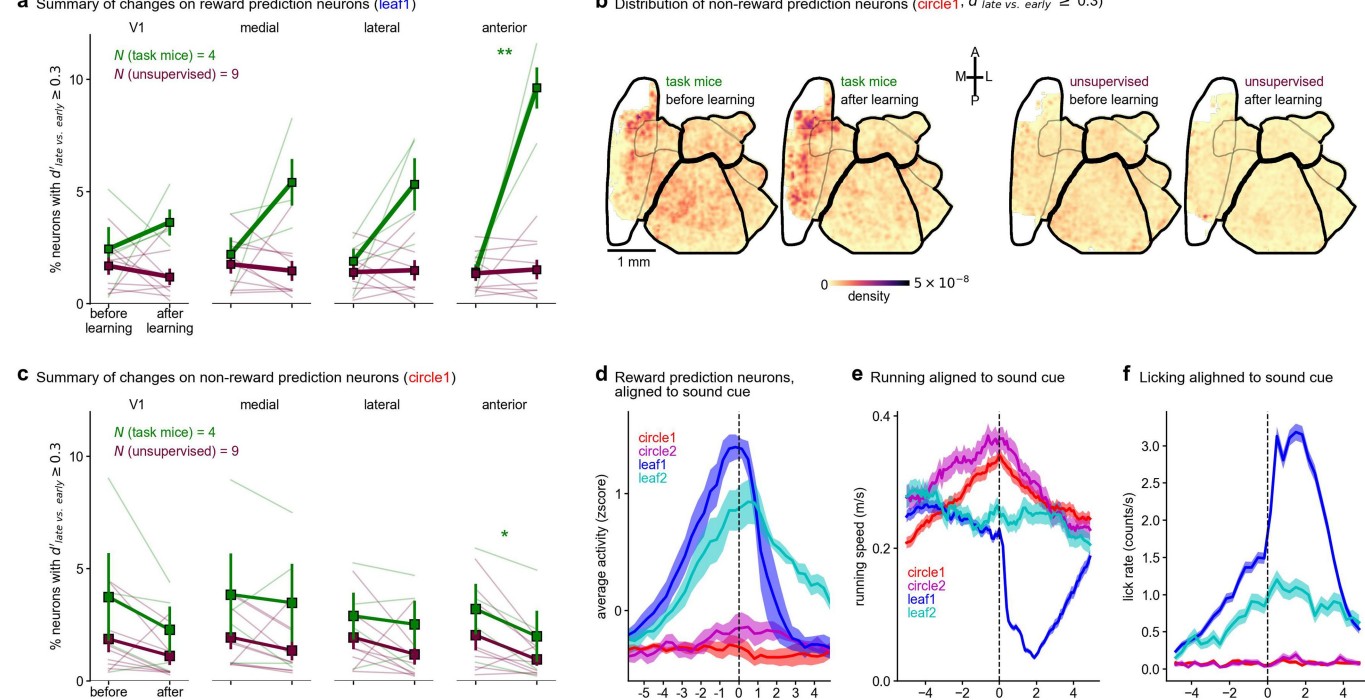

**Extended Data Fig. 8 | Reward and non-reward prediction neurons across areas. a**, (left) Percentage of reward-selective neurons before and after learning across all regions. (right) Increase in fraction of selective neurons (task mice: n=4 mice; unsupervised: n=9 mice). **b**, Distribution of non-reward prediction neurons, defined as neurons with a $d'_{\text{late vs early}} \geq 0.3$ computed from the circle1 corridor trials, which are non-rewarding. **c**, Summary of changes from before to after learning for the non-reward prediction neurons. **d**, Reward prediction neuron average response aligned to the sound cue for all four corridors in the *test1* session. **e,f**, Running speed and licking rate respectively, aligned to sound cue. All data are mean ± s.e.m. *$P$<0.05, **$P$<0.01.

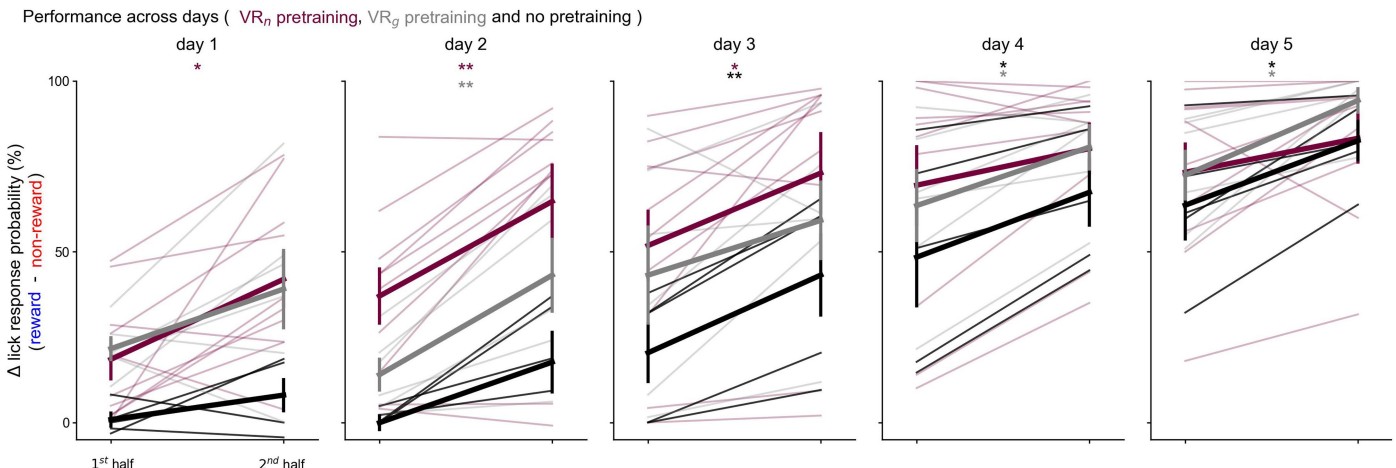

**Extended Data Fig. 9 | Within-day learning in unsupervised pretraining experiment.** Behavioral performance over days, split into the first and second halves of the session (n=11 VR$_n$ pretraining mice, n=7 VR$_g$ pretraining mice, n=5 no pretraining mice). All data are mean ± s.e.m. *$P$<0.05, **$P$<0.01.

# Reporting Summary

## Statistics

For all statistical analyses, confirm that the following items are present in the figure legend, table legend, main text, or Methods section.

| n/a | Confirmed | |
|---|---|---|
| ☐ | ☒ | The exact sample size (*n*) for each experimental group/condition, given as a discrete number and unit of measurement |
| ☐ | ☒ | A statement on whether measurements were taken from distinct samples or whether the same sample was measured repeatedly |
| ☐ | ☒ | The statistical test(s) used AND whether they are one- or two-sided<br>*Only common tests should be described solely by name; describe more complex techniques in the Methods section.* |
| ☒ | ☐ | A description of all covariates tested |
| ☒ | ☐ | A description of any assumptions or corrections, such as tests of normality and adjustment for multiple comparisons |
| ☐ | ☒ | A full description of the statistical parameters including central tendency (e.g. means) or other basic estimates (e.g. regression coefficient) AND variation (e.g. standard deviation) or associated estimates of uncertainty (e.g. confidence intervals) |
| ☐ | ☒ | For null hypothesis testing, the test statistic (e.g. *F*, *t*, *r*) with confidence intervals, effect sizes, degrees of freedom and *P* value noted<br>*Give P values as exact values whenever suitable.* |
| ☒ | ☐ | For Bayesian analysis, information on the choice of priors and Markov chain Monte Carlo settings |
| ☒ | ☐ | For hierarchical and complex designs, identification of the appropriate level for tests and full reporting of outcomes |
| ☐ | ☒ | Estimates of effect sizes (e.g. Cohen's *d*, Pearson's *r*), indicating how they were calculated |

*Our web collection on statistics for biologists contains articles on many of the points above.*

## Software and code

Policy information about availability of computer code

| Data collection | Matlab 2022, psychtoolbox 3.0.19 |
|---|---|
| Data analysis | python 3.12.8, numpy 2.0., scipy 1.14.1, matplotlib 3.10.0, suite2p 0.14.4, rastermap 0.1.3, suite2p 0.14.2 |

For manuscripts utilizing custom algorithms or software that are central to the research but not yet described in published literature, software must be made available to editors and reviewers. We strongly encourage code deposition in a community repository (e.g. GitHub). See the Nature Portfolio guidelines for submitting code & software for further information.

## Data

Policy information about availability of data

All manuscripts must include a data availability statement. This statement should provide the following information, where applicable:
- Accession codes, unique identifiers, or web links for publicly available datasets
- A description of any restrictions on data availability
- For clinical datasets or third party data, please ensure that the statement adheres to our policy

Data is now available at  https://doi.org/10.25378/janelia.28811129.v1
Accompanying code to reproduce analyses in the paper is available at https://github.com/MouseLand/zhong-et-al-2025.

# Research involving human participants, their data, or biological material

Policy information about studies with human participants or human data. See also policy information about sex, gender (identity/presentation), and sexual orientation and race, ethnicity and racism.

| | |
|---|---|
| Reporting on sex and gender | n/a |
| Reporting on race, ethnicity, or other socially relevant groupings | n/a |
| Population characteristics | n/a |
| Recruitment | n/a |
| Ethics oversight | n/a |

Note that full information on the approval of the study protocol must also be provided in the manuscript.

# Field-specific reporting

Please select the one below that is the best fit for your research. If you are not sure, read the appropriate sections before making your selection.

☒ Life sciences ☐ Behavioural & social sciences ☐ Ecological, evolutionary & environmental sciences

For a reference copy of the document with all sections, see nature.com/documents/nr-reporting-summary-flat.pdf

# Life sciences study design

All studies must disclose on these points even when the disclosure is negative.

| | |
|---|---|
| Sample size | No sample size calculation was performed. We used 3-9 mice for each group, which is common for neural recordings with behavior, and sufficient due to recording a very large number of neurons per mouse (~50,000). In cases where 3 mice were used (naive group), multiple sessions were repeated with independent stimulus sets and treated as different samples. |
| Data exclusions | No data was excluded. |
| Replication | Each mouse trained and recorded can be viewed as a replication of the experiment. Each mouse generated a dataset of about 50,000 neurons per session, and all the major results held in each mouse individually. |
| Randomization | Allocation was random. |
| Blinding | Investigators were not blinded, because the supervised cohort of mice had to be water restricted and trained to produce the behavioral choices. |

# Reporting for specific materials, systems and methods

We require information from authors about some types of materials, experimental systems and methods used in many studies. Here, indicate whether each material, system or method listed is relevant to your study. If you are not sure if a list item applies to your research, read the appropriate section before selecting a response.

## Materials & experimental systems

| n/a | Involved in the study |
|---|---|
| ☒ | Antibodies |
| ☒ | Eukaryotic cell lines |
| ☒ | Palaeontology and archaeology |
| ☐ | ☒ Animals and other organisms |
| ☒ | Clinical data |
| ☒ | Dual use research of concern |
| ☒ | Plants |

## Methods

| n/a | Involved in the study |
|---|---|
| ☒ | ChIP-seq |
| ☒ | Flow cytometry |
| ☒ | MRI-based neuroimaging |

# Animals and other research organisms

| | |
|---|---|
| Laboratory animals | The study involved C57 mice as well as gcamp6s transgenic mice (tetO-gcamp6s x Camk2a-tTA), both male and female, at 2-11 months of age. The mice were housed in reversed light cycle in individually ventilated cages kept at 46-57% humidity and 21-22C. |
| Wild animals | No wild animals were used in this study. |
| Reporting on sex | Both female and male mice were used, but we could analyze data by subgroup because the experiments are labor-intensive and not enough animals were recorded. |
| Field-collected samples | No field samples were used in this study. |
| Ethics oversight | All experimental procedures were conducted according to IACUC, and received ethical approval from the IACUC board at HHMI Janelia Research Campus. |

Note that full information on the approval of the study protocol must also be provided in the manuscript.

# Plants

| | |
|---|---|
| Seed stocks | N/A |
| Novel plant genotypes | N/A |
| Authentication | N/A |

