## [Peer Review File · Nature]

Unsupervised pretraining in biological neural networks

Corresponding Author: Dr Marius Pachitariu

Version 0:

Reviewer comments:

Referee #1

(Remarks to the Author)

In a series of head-fixed experiments, Zhong et al. demonstrate neural changes across mouse visual cortex in response to different versions of an immersive visual discrimination task. Neural changes found in “task mice” (where mice were rewarded with a drop of water) were replicated in mice who had unrewarded exposure to the same visual stimuli. The only exception were neurons in anterior HVAs, where a ramping reward prediction signal was found. Unsupervised pre-training further accelerated subsequent task learning.

The authors tackle an important and open question in systems neuroscience: to what extent are neural changes induced by plasticity due to supervised vs unsupervised learning? While the literature on supervised (i.e., reward-based) learning is rich, the authors claim that only indirect evidence exists of unsupervised learning. However, the manuscript does not mention the literature on Hebbian/anti-Hebbian plasticity and related forms of unsupervised synaptic plasticity. Still, the finding that both experimental conditions (with and without an extrinsic reward) lead to neural changes remains novel and (to my knowledge) unprecedented at this scale and detail.

Methods are rigorous and cutting-edge. The system and procedure are described in sufficient detail in the Methods and Appendices. Both the quality of the data and the quality of the presentation are top notch.

Where I take issue is with the conclusions. While the results are impressive, they do not demonstrate “distinct streams” of supervised and unsupervised learning. In fact, the word “stream” only appears in the title and in Fig. 6i. Certainly, reward-based neural changes seem to be restricted to anterior regions, and this is consistent with previous findings. But there was no evidence presented that 1) this reward-prediction signal was the driving factor behind neural changes and that 2) this is unrelated to the neural changes seen in other areas.

More detailed comments:

Abstract: I think here “feedback” refers to the teacher signal common in supervised learning, but in a neuroscience context could be misunderstood as “top-down connections”

I.11: What about STDP and related unsupervised methods?

I.433: How many mice were male vs female?

I.444: How many C57 mice were male vs female?

A general comment is that the orange vs green choice in figure plots is not very colorblind-friendly.

Referee #2

(Remarks to the Author)

In this study, Zhong et al. examine the impact of both unsupervised and reinforced experience with visual stimuli on neural

representations and reinforcement learning. The authors conducted simultaneous recordings of populations of neurons, numbering up to 90,000, from both the primary visual cortex (V1) and higher visual areas (HVA) in mice. This was done while the mice were exposed to the same stimuli with and without any reward. Performance in mice that received rewards was measured through anticipatory licking to the correct stimuli. Consistent with prior research, the authors observed that changes in neural activity in reward receiving mice were linked to their learning progress. But, similar neural alterations were evident in mice exposed to the stimuli without reward. In particular, the authors saw orthogonalization of the representations for different stimuli, regardless of whether the animals received rewards or not. Interestingly, animals that were preexposed to the stimuli without rewards also then showed faster learning when given rewards, suggesting that this orthogonalization process may help with downstream task learning. Additionally, in mice doing reinforcement learning, the authors identified a gradual increase in responses related to reward prediction, most prominently in the anterior HVAs.

Overall, I very much enjoyed this paper and believe it makes a very important contribution to the field. Theorists have long postulated that unsupervised exposure to stimuli should help animals with downstream learning, and there is evidence for this hypothesis dating back many decades, including both behavioural evidence (see e.g. Gibson & Walk, 1956: <https://psycnet.apa.org/doi/10.1037/h0048274>) and neural evidence (e.g. Blakemore & Cooper, 1970: <https://doi.org/10.1038/228477a0>). But, no studies have explored the two issues simultaneously that I am aware of, and certainly not with the ability to measure such large populations of neurons across so many areas. As such, I think this will be a high impact study that is worthy of publication in Nature.

But, before it is ready for publication, I think there are two major issues with the paper in its current form that need to be addressed. I describe these below, then highlight some more minor issues that I also think the authors should attend to.

Major concern #1

There is one important control in the unsupervised pretraining experiments that is missing, and which prevents me from saying that the data clearly shows an impact from unsupervised learning on performance. Specifically, there is no control for the amount of time spent in the VR system. Mice that undergo unsupervised pretraining are getting 10 extra days in the VR. What if their improved performance is a result of this increased exposure to the experimental apparatus? Moreover, the current experiments do not conclusively show that it is exposure to the relevant stimuli that drives the improved performance.

To control for these issues, the authors should run an experiment wherein a group of mice undergo the same 10 days in the 'VR only' condition as the pretrained mice, but they should be exposed to totally unrelated visual stimuli. For example, if the mice are presented with highly artificial stimuli that have very different spectral components (such as gratings or checkerboards) we would hypothesize that they would not learn any useful information to help them distinguish the circle and leaf stimuli. In that case, if it is truly an unsupervised learning effect, we would not predict any performance boosts. If, on the other hand, we did see performance boosts, that would imply that it was not learning about the stimuli, per se, that helped, but rather, increased exposure to the apparatus.

Note also that this would allow the authors to test for the orthogonalization effects and correlate it with any behavioural improvements. If you see (1) improved performance in only the animals pretrained on circle/leaf (and not grating/checkerboard, say), (2) orthogonalization of the circle/leaf stimuli in the circle/leaf group, but (3) no leaf/circle orthogonalization in a grating/checkerboard group, then it is much more reasonable to claim that the performance boost is a result of the orthogonalization effects.

I believe that these controls are critical for supporting the final conclusions of the paper, and I believe that they do not present too large a burden in new experiments, given that it wouldn't require any change to the experimental apparatus or training procedures. Thus, I believe that these experiments are necessary for publication in Nature.

Major concern #2

I think the authors are overstating things when they argue that the reward signals are "anterior", and I think their final cartoon (Fig. 6i) is too simplistic and misleading. Yes, there was a higher density of reward signals in the anterior regions, but Figure S5 shows there was a trend towards an increase in reward selective neurons in all regions (I suspect the lack of statistical significance is just due to a small n). Similarly, it is clear from figures 2-4 that novelty detection and orthogonalization also occur almost everywhere, just to differing degrees.

As such, I think that final cartoon needs to be dropped or altered and the authors need to change the manuscript to reflect the reality that their data presents: the learning related changes that occur are widely distributed and every region seems to be participating to some degree in all of the facets of learning, but to different degrees. Even just modifying the final cartoon to be a colour coded mixture of the different functions would be better, Without these changes I think many non-careful readers will come out of this paper with a really mistaken understanding of what the data shows, thinking that there was a really clean anatomical split between these functions. That would be a real shame for an otherwise wonderful paper.

Minor comments

- This is just a point of terminology, but “supervised” learning is often reserved for situations where there is an explicit target given for the output activity. In contrast, here, the mice are undergoing “reinforcement” learning. I would strongly recommend, for sake of not confusing readers, switching to the terminology “reinforced” versus “unsupervised”, rather than “supervised” versus “unsupervised”.

- Lines 17-18: technically, “predictive coding” as originally envisioned by Rao and Ballard refers to an encoding scheme wherein neurons represent predictions and sensory prediction errors, and only the errors are communicated downstream, while the predictions are communicated upstream. It needn’t even involve learning, per se. Predictive *learning* (not coding) algorithms are a much broader category of learning, and do not necessarily use predictive coding. These include older models as well, such as Helmholtz machines, etc.

- Lines 278-281: why do you use late versus early reward to identify reward selective neurons? The logic was unclear to me from the text.

- It was not clear from the results or methods how the rock and brick stimuli were used. Were these stimuli shown to all mice, or just those in the unsupervised pretraining tests? More information is needed for the reader.

- The authors should consider citing the classic papers I highlighted above, in order to recognize that their study is adding to existing knowledge in this area.

Referee #3

(Remarks to the Author)

The study from Zhong et al asks whether existing results relating learning-induced changes in visual responses may be accounted for by simple exposure to stimuli without reward association.

The experimental approach involves an impressive survey of multiple visual cortical areas at single cell resolution. The dataset thus generated may prove very valuable. My enthusiasm for this study and its conclusions is unfortunately reduced due to issues with the experimental design, analysis and the importance of the conclusions.

The behaviour design is unfortunately poorly suited to answer this question. Ideally, one would like to study the responses to the same cleanly presented stimuli with comparable overt behaviour, pre vs post learning/exposure. Some sacrifices to this ideal are usually justifiable. However, in this VR based approach, visual stimuli are not repeatable but depend on the movement of the mice through the corridor. While there may be ways to deal with this in analysis, simply removing stationary epochs is not sufficient. The mice are presented with a reward randomly anywhere in the rewarded corridor, and so the mice lick continuously in the rewarded corridors. The average response summed up over the entire corridor is thus a mixture of multiple factors including reward and anticipation responses. Critically, what are the mice actually recognising? Clearly it is the first view of the rewarded (or unrewarded) corridor which becomes associated with reward (or non reward), since it guarantees reward in a random upcoming location. Further visual stimuli in the rewarded corridor are not providing any new information about reward. This has not been considered in this design. Even re-analysis of this data will not allow a clean study of this response since this stimulus does not appear abruptly but gradually appears from the horizon. This prevents knowing when the mice were first presented with the rewarded (or unrewarded) corridor (first lick would be too late). And this moment is the key ‘learnt’ stimulus.

The analysis also has some issues. For unclear reasons the authors use arbitrary cutoffs ($d' > 0.3$ etc), and rely heavily on that to get cell densities etc. Critically, very little analysis of overt behaviour has been included, particularly relating to how much various overt movements account for neural responses.

The overall novelty of their claim seems to be that other studies have got it wrong, since previous studies did not do good stimulus exposure-only controls. I believe there is some truth to pointing out this gap in the existing knowledge. However, regarding V1 studies (most well studied) there have been numerous studies that have found increased stimulus selectivity with learning, and this study has found no increase in V1 selectivity, and also not in the exposure only control. Thus, given that the authors have not been able to replicate the earlier findings in the first place, and given that this is likely due to the numerous choices of experimental design and analysis mentioned earlier, their major claim becomes difficult to believe.

A major distinction made is that “the medial population did not represent trial-to-trial variability in mouse decision-making, while anterior HVAs did”. However, the difference between the learning and unsupervised group is the key measure here. This is the same between the medial and anterior areas.

Other claims include identifying a novelty response in V1, but as they mentioned themselves, this is not a novel finding. The improved learning rate after pre-exposure is also interesting but not novel.

Minor: There is some phrasing which causes confusion regarding stimulus responses vs percentage of cells above a threshold of selectivity. Eg. “The responses of leaf2-selective neurons adapted substantially after an additional week of training with the leaf2 stimulus (Figure 3a,b).” It will be important to revise all wording about “responses”, and maybe even include more analysis about response amplitudes (to the same visual stimulus) beyond the existing.

Version 1:

Reviewer comments:

Referee #1

(Remarks to the Author)

The authors have addressed all of my concerns.

Referee #2

(Remarks to the Author)

The authors have done an excellent job of addressing my concerns, and I believe the updated paper should be accepted for publication.

I still have a quibble with the authors' use of the term "supervised" for their task with reward, and I don't agree with them that reinforcement learning implies a lack of perceptual learning. As the authors noted, many RL models do involve perceptual learning. In fact almost *all* modern RL models in AI involve learning sensory representations. Only toy models do not.

Added to this, I don't think the animals actually receive a supervisory signal simply because they are being asked to engage in a simple behavior (licking). Importantly, the animals do not receive a literal signal of which neurons to activate at which time (which is what is typically what is meant by "supervised" learning, at least in AI).

But, such a semantic quibble is not something that should prevent the paper from being published. I will simply leave these comments here for the authors to consider in their final revision.

Referee #3

(Remarks to the Author)

While the manuscript is substantially improved in certain aspects, I remain sceptical of some key issues.

Overall, I think this manuscript does provide useful insights into the question of unsupervised exposure and how this allows subsequent improvements in supervised learning. However, I still have a concern about the generalisability of the unsupervised results, see below, and also, as mentioned by the authors and other reviewers, it is not entirely novel that prior exposure improves learning, nor that it changes neural representations, but this study puts the two together. While this is indeed an important contribution, and the orthogonalization is an interesting result, in my opinion this would not on its own be suitable for this journal.

Which is why this study goes further and makes key claims related to the comparison of unsupervised to supervised learning. Here is where I am still not convinced that the data is supporting the claims.

I have 3 broad concerns remaining:

1. I still don't think the authors are necessarily studying the relevant learning-related changes in neural responses. The authors point out their experiment with swapping the corridor in half. While this is good evidence that the mice do associate more than the start of the corridor with reward, it is still the case that the mice will most prominently learn the first view of the rewarded corridor, following which the reward is guaranteed (the lick rasters and response profiles should ideally be extended to earlier time points than 0 to allow readers to get a good sense of how early they start licking/responding). I don't believe it is clear what the difference is between this first-view-learning and subsequent exposure while the animal is licking and waiting for its guaranteed reward. There is the additional confound that the earlier parts of the corridor will have many more bouts of pre-reward licking, vs later corridor parts. Overall, while the author's claim is indeed *plausible*, I would not consider it *strong evidence* that they have characterised the relevant learning related neural changes, and thus have made a fair comparison with the unsupervised cohort.
2. Confounding responses to overt behaviour: The authors respond that since they find no difference between learning and unsupervised cohorts, this means concerns about contamination of responses with overt behaviour is not a concern. I agree that this is also a plausible argument, but I would argue that unless behavioural responses are not dealt with carefully, we are essentially dealing with poorer quality data, and there may be unexpected effects cancelling each other out. Perhaps the currently visible but subtle pattern of differences within a brain region between the cohorts (eg Fig 1i) will get enhanced, or perhaps the claim of the authors is made even stronger. Again, the claim as it stands is plausible but not the highest quality of data expected.
3. Taking the current results at face value, there are still some inconsistencies. First, a previous point of mine was misunderstood about the differences between supervised and unsupervised cohorts. I was referring to Fig 1j, 'medial' category, post learning, the data is as different as in the 'anterior' category. This suggests that medial and anterior both show a stronger effect of supervised compared to unsupervised (could be tested more directly by testing for an interaction effect on the pair of lines). Thus, there are more differences between unsupervised and supervised than suggested. Second, the new results with gratings, Supp Fig 1d, shows almost no changes before vs after exposure, in contrast to Fig 1i with natural stimuli. This raises doubts about how general the findings of the unsupervised exposure are.

Referee #1 (Remarks to the Author):

In a series of head-fixed experiments, Zhong et al. demonstrate neural changes across mouse visual cortex in response to different versions of an immersive visual discrimination task. Neural changes found in “task mice” (where mice were rewarded with a drop of water) were replicated in mice who had unrewarded exposure to the same visual stimuli. The only exception were neurons in anterior HVAs, where a ramping reward prediction signal was found. Unsupervised pre-training further accelerated subsequent task learning.

The authors tackle an important and open question in systems neuroscience: to what extent are neural changes induced by plasticity due to supervised vs unsupervised learning? While the literature on supervised (i.e., reward-based) learning is rich, the authors claim that only indirect evidence exists of unsupervised learning. However, the manuscript does not mention the literature on Hebbian/anti-Hebbian plasticity and related forms of unsupervised synaptic plasticity. Still, the finding that both experimental conditions (with and without an extrinsic reward) lead to neural changes remains novel and (to my knowledge) unprecedented at this scale and detail.

Methods are rigorous and cutting-edge. The system and procedure are described in sufficient detail in the Methods and Appendices. Both the quality of the data and the quality of the presentation are top notch.

Where I take issue is with the conclusions. While the results are impressive, they do not demonstrate “distinct streams” of supervised and unsupervised learning. In fact, the word “stream” only appears in the title and in Fig. 6i. Certainly, reward-based neural changes seem to be restricted to anterior regions, and this is consistent with previous findings. But there was no evidence presented that 1) this reward-prediction signal was the driving factor behind neural changes and that 2) this is unrelated to the neural changes seen in other areas.

Thank you for the positive review. In the revision, we rectified the omission of the literature on unsupervised synaptic plasticity such as Hebbian/anti-Hebbian plasticity (Lines 26-30) and we also added a sentence to the discussion (Lines 448-453). We have also changed the title from “Distinct streams...” to “Unsupervised pretraining in biological neural networks” to more closely align with the results we presented. We removed the cartoon showing “distinct streams” for supervised and unsupervised learning.

More detailed comments:

Abstract: I think here “feedback” refers to the teacher signal common in supervised learning, but in a neuroscience context could be misunderstood as “top-down connections”

Thanks, we modified this to “instruction”.

I.11: What about STDP and related unsupervised methods?

As mentioned above, we added mentions of synaptic plasticity mechanisms in the introduction and discussion (Lines 26-30, 448-453).

I.433: How many mice were male vs female?

For the imaging experiments, 6 mice were female and 13 mice were male. We added this information in the methods (Lines 496-497).

I.444: How many C57 mice were male vs female?

The C57 mice were used for the behavior-only task. All of them were female. We added this information in the methods (Line 511).

A general comment is that the orange vs green choice in figure plots is not very colorblind-friendly.

Thanks for pointing that out, we changed the color scheme to green vs purple.

Referee #2 (Remarks to the Author):

In this study, Zhong et al. examine the impact of both unsupervised and reinforced experience with visual stimuli on neural representations and reinforcement learning. The authors conducted simultaneous recordings of populations of neurons, numbering up to 90,000, from both the primary visual cortex (V1) and higher visual areas (HVA) in mice. This was done while the mice were exposed to the same stimuli with and without any reward. Performance in mice that received rewards was measured through anticipatory licking to the correct stimuli. Consistent with prior research, the authors observed that changes in neural activity in reward receiving mice were linked to their learning progress. But, similar neural alterations were evident in mice exposed to the stimuli without reward. In particular, the authors saw orthogonalization of the representations for different stimuli, regardless of whether the animals received rewards or not. Interestingly, animals that were preexposed to the stimuli without rewards also then showed faster learning when given rewards, suggesting that this orthogonalization process may help with downstream task learning. Additionally, in mice doing reinforcement learning, the authors identified a gradual increase in responses related to reward prediction, most prominently in the anterior HVAs.

Overall, I very much enjoyed this paper and believe it makes a very important contribution to the field. Theorists have long postulated that unsupervised exposure to stimuli should help animals with downstream learning, and there is evidence for this hypothesis dating back many

decades, including both behavioural evidence (see e.g. Gibson & Walk, 1956: <https://psycnet.apa.org/doi/10.1037/h0048274>) and neural evidence (e.g. Blakemore & Cooper, 1970: <https://doi.org/10.1038/228477a0>). But, no studies have explored the two issues simultaneously that I am aware of, and certainly not with the ability to measure such large populations of neurons across so many areas. As such, I think this will be a high impact study that is worthy of publication in Nature.

But, before it is ready for publication, I think there are two major issues with the paper in its current form that need to be addressed. I describe these below, then highlight some more minor issues that I also think the authors should attend to.

Thank you for the positive review. We added the references mentioned above, and we addressed the major concerns below with substantial new experiments.

Major concern #1

There is one important control in the unsupervised pretraining experiments that is missing, and which prevents me from saying that the data clearly shows an impact from unsupervised learning on performance. Specifically, there is no control for the amount of time spent in the VR system. Mice that undergo unsupervised pretraining are getting 10 extra days in the VR. What if their improved performance is a result of this increased exposure to the experimental apparatus? Moreover, the current experiments do not conclusively show that it is exposure to the relevant stimuli that drives the improved performance.

To control for these issues, the authors should run an experiment wherein a group of mice undergo the same 10 days in the 'VR only' condition as the pretrained mice, but they should be exposed to totally unrelated visual stimuli. For example, if the mice are presented with highly artificial stimuli that have very different spectral components (such as gratings or checkerboards) we would hypothesize that they would not learn any useful information to help them distinguish the circle and leaf stimuli. In that case, if it is truly an unsupervised learning effect, we would not predict any performance boosts. If, on the other hand, we did see performance boosts, that would imply that it was not learning about the stimuli, per se, that helped, but rather, increased exposure to the apparatus.

Thank you for this suggestion for an important control. We did exactly this, and habituated 7 C57 mice to grating stimuli presented in the VR setting for the same amount of time as the naturalistic textures were shown in the existing "unsupervised" cohort (Figure 6). The grating-exposure mice learned as fast as the no-exposure mice, and thus we conclude that learning about the stimulus helped the unsupervised cohort. Note that in the revised figures we separated the first day of training (now "day 0"), since rewards were given passively on that day without requiring licks and thus licking cannot be taken as an indication of stimulus learning (Lines 379-381). The new control is described in Lines 373-376 and figure 6. We also added 4 more mice to the unsupervised cohort of pretrained mice exposed to naturalistic textures as an

additional matched control. We observed no differences compared to the original 7 pretrained mice and thus we pooled both groups together for a total of 11 mice in the pretrained group.

Note also that this would allow the authors to test for the orthogonalization effects and correlate it with any behavioural improvements. If you see (1) improved performance in only the animals pretrained on circle/leaf (and not grating/checkerboard, say), (2) orthogonalization of the circle/leaf stimuli in the circle/leaf group, but (3) no leaf/circle orthogonalization in a grating/checkerboard group, then it is much more reasonable to claim that the performance boost is a result of the orthogonalization effects.

Thanks for the suggestions. Point (1) is now demonstrated by the new grating-exposure cohort in figure 6. Point (2) we believe refers to the plasticity we reported in the original paper. Point (3) required another set of new experiments, because the grating-only mice from the new behavioral experiments undergo a different training protocol compared to the imaging mice. To demonstrate point (3), we therefore ran another set of experiments in *gcamp6* transgenic mice with imaging. We indeed found no leaf/circle plasticity (including orthogonalization) in the grating group. The new control analyses are presented in Figures 1j, 3deh and 4d.

I believe that these controls are critical for supporting the final conclusions of the paper, and I believe that they do not present too large a burden in new experiments, given that it wouldn't require any change to the experimental apparatus or training procedures. Thus, I believe that these experiments are necessary for publication in Nature.

Thank you, we believe the two new sets of experiments address this in full.

Major concern #2

I think the authors are overstating things when they argue that the reward signals are “anterior”, and I think their final cartoon (Fig. 6i) is too simplistic and misleading. Yes, there was a higher density of reward signals in the anterior regions, but Figure S5 shows there was a trend towards an increase in reward selective neurons in all regions (I suspect the lack of statistical significance is just due to a small *n*). Similarly, it is clear from figures 2-4 that novelty detection and orthogonalization also occur almost everywhere, just to differing degrees.

As such, I think that final cartoon needs to be dropped or altered and the authors need to change the manuscript to reflect the reality that their data presents: the learning related changes that occur are widely distributed and every region seems to be participating to some degree in all of the facets of learning, but to different degrees. Even just modifying the final cartoon to be a colour coded mixture of the different functions would be better, Without these changes I think many non-careful readers will come out of this paper with a really mistaken understanding of what the data shows, thinking that there was a really clean anatomical split between these functions. That would be a real shame for an otherwise wonderful paper.

Thanks for the suggestion, we agree with this. We dropped the cartoon and changed the title of the paper from “Distinct streams...” to “Unsupervised pretraining in biological neural networks”. We believe the new title more closely aligns with what we actually showed and addresses the concerns of the reviewer.

Minor comments

- This is just a point of terminology, but “supervised” learning is often reserved for situations where there is an explicit target given for the output activity. In contrast, here, the mice are undergoing “reinforcement” learning. I would strongly recommend, for sake of not confusing readers, switching to the terminology “reinforced” versus “unsupervised”, rather than “supervised” versus “unsupervised”.

This is a good point, and we ourselves go back and forth between “reinforced” and “supervised”. The problem with “reinforced” is that it immediately suggests “reinforcement learning”, and contrasting that with unsupervised learning would be the wrong comparison here. Reinforcement learning is a third category of learning algorithms that deals with high-dimensional action-value-state associations. There is no perceptual learning usually in these problems (exception being some of the DeepMind algorithms that learn a vision model as well).

We do not think the term supervised learning is inaccurate in our case, because an animal’s choice on each trial is the output of its neural network, and the correct label is indicated via reward. It is true that the task also has some reinforcement learning components, but these did not in fact seem to result in differences in the neural plasticity compared to the unsupervised cohort, except in the anterior area. We added this additional emphasis in the text (Lines 130-140). We do have some indications that the reinforcement learning component is not different for example between our supervised and unsupervised cohorts in figure 6, because they learn the position of the reward at similar rates (Figure 6g).

- Lines 17-18: technically, “predictive coding” as originally envisioned by Rao and Ballard refers to an encoding scheme wherein neurons represent predictions and sensory prediction errors, and only the errors are communicated downstream, while the predictions are communicated upstream. It needn’t even involve learning, per se. Predictive *learning* (not coding) algorithms are a much broader category of learning, and do not necessarily use predictive coding. These include older models as well, such as Helmholtz machines, etc.

We made this statement more precise as: “...have been interpreted as evidence for predictive coding **arising from** unsupervised learning” (Lines 20-21. We are already citing Helmholtz machines (i.e. Hinton et al 1995).

- Lines 278-281: why do you use late versus early reward to identify reward selective neurons? The logic was unclear to me from the text.

This was done to control for the effect of stimulus drive and reward-related neural activity. The stimuli are the same in early- vs late-reward trials when averaged over all timepoints. In contrast, the anticipatory neural activity is more prolonged in the late-reward trials. There could have been other ways to pick these neurons, such as making an encoding model with the relevant variables, but we thought the early- vs late-reward trials have a built-in control that substantially simplifies this analysis. We clarified this choice in the text (Lines 318-324).

- It was not clear from the results or methods how the rock and brick stimuli were used. Were these stimuli shown to all mice, or just those in the unsupervised pretraining tests? More information is needed for the reader.

Most of the imaging mice saw leaf/circle, but in three mice we also used rock, brick or circle for rewarded stimulus. We did not have enough mice to show this as a full control with statistics, but the trends were the same. We added a mention in the text to make this more clear (Lines 53-54).

- The authors should consider citing the classic papers I highlighted above, in order to recognize that their study is adding to existing knowledge in this area.

Thanks, we added the two classical studies mentioned.

Referee #3 (Remarks to the Author):

The study from Zhong et al asks whether existing results relating learning-induced changes in visual responses may be accounted for by simple exposure to stimuli without reward association.

The experimental approach involves an impressive survey of multiple visual cortical areas at single cell resolution. The dataset thus generated may prove very valuable. My enthusiasm for this study and its conclusions is unfortunately reduced due to issues with the experimental design, analysis and the importance of the conclusions.

We thank the reviewer for their comments, which we address below.

The behaviour design is unfortunately poorly suited to answer this question. Ideally, one would

like to study the responses to the same cleanly presented stimuli with comparable overt behaviour, pre vs post learning/exposure.

See below why we think this behavior was a good choice more generally especially in the context of the existing literature, but specifically to these points, we need to emphasize some details of the task and behavior. First, the speed of the VR is held fixed when the animals are running, which means they all get more or less the same visual experience, except for the times they stop running, and we do not consider those timepoints for any analyses. Second, the running speeds, although they do not control the VR speed, are in fact very similar both before and after learning, and between supervised and unsupervised cohorts. We added these details in Lines 78-86 and new Figure S2.

That said, our results do not depend strongly on whether these behaviors are similar between cohorts, because we mostly show similarities, not differences in neural activity between the supervised and unsupervised cohorts. If we had shown differences, one could indeed be a lot more worried that those differences in fact arise from behavioral differences. This is why for the only analysis where we do find differences (Figure 5), we ensure that the “special” class of neurons we found is not due to rewards or behavior, by controlling for those with our selection criterion between early-reward trials and far-reward trials. Similarly, we control for licking by aligning for the first lick in the corridors, and find that the neural ramping activity precedes it by several seconds. We added an explicit statement of these motivations in Lines 318-324.

Some sacrifices to this ideal are usually justifiable. However, in this VR based approach, visual stimuli are not repeatable but depend on the movement of the mice through the corridor. While there may be ways to deal with this in analysis, simply removing stationary epochs is not sufficient.

Sorry for the confusion here, this is actually not true: stimuli were moving at a fixed speed when the mice ran above a threshold. This was only stated in the methods, so we moved this important information to the main text (Lines 78-81).

The mice are presented with a reward randomly anywhere in the rewarded corridor, and so the mice lick continuously in the rewarded corridors. The average response summed up over the entire corridor is thus a mixture of multiple factors including reward and anticipation responses.

This is also not entirely true. As stated in the main text, only running timepoints are considered for analysis, which removes all reward and consummatory periods (several seconds). The activity is also not summed over timepoints, but rather treated as distribution, pooled over timepoints, and compared between corridors (Lines 92-96). Mice sometimes do lick in anticipation before water delivery, and *despite* this fact, the patterns of neural plasticity were very similar between the supervised and unsupervised cohorts. As we show in Figure 5, this is because most of the activity related to the anticipatory state is restricted to the anterior HVAs.

Critically, what are the mice actually recognising? Clearly it is the first view of the rewarded (or unrewarded) corridor which becomes associated with reward (or non reward), since it

guarantees reward in a random upcoming location. Further visual stimuli in the rewarded corridor are not providing any new information about reward. This has not been considered in this design. Even re-analysis of this data will not allow a clean study of this response since this stimulus does not appear abruptly but gradually appears from the horizon. This prevents knowing when the mice were first presented with the rewarded (or unrewarded) corridor (first lick would be too late). And this moment is the key 'learnt' stimulus.

This is an interesting hypothesis, but we did already have analyses to suggest this is not the case. Specifically in Figure 4fg we showed that when the corridor images are swapped between the first and second half, or first quarter and next three quarters, the behavior remains the same. This suggests the mice do not in fact only use the beginning of the corridor for discrimination / learning.

The analysis also has some issues. For unclear reasons the authors use arbitrary cutoffs ($d' > 0.3$ etc), and rely heavily on that to get cell densities etc.

We had to choose some value of the cutoff for the analyses. We added a supplementary figure to show that the changes do not depend on the precise value of the cutoff (Figure S1e, Lines 113-114).

Critically, very little analysis of overt behaviour has been included, particularly relating to how much various overt movements account for neural responses.

We added a supplementary figure to show that the running behaviors were very similar before and after learning, in the supervised and unsupervised cohorts (Figure S2, Lines 81-84). We want to emphasize again that even if there had been differences in overt movements, the similarity between supervised and unsupervised plasticity would still be true *despite* those differences. This similarity is the main point of the paper.

The overall novelty of their claim seems to be that other studies have got it wrong, since previous studies did not do good stimulus exposure-only controls. I believe there is some truth to pointing out this gap in the existing knowledge. However, regarding V1 studies (most well studied) there have been numerous studies that have found increased stimulus selectivity with learning, and this study has found no increase in V1 selectivity, and also not in the exposure only control. Thus, given that the authors have not been able to replicate the earlier findings in the first place, and given that this is likely due to the numerous choices of experimental design and analysis mentioned earlier, their major claim becomes difficult to believe.

We agree that this is an interesting observation. There are two things to consider:

1) We did find changes in V1 tuning after learning, see Figures 3 and 4. What we did not find were changes in the *number* of selective neurons, which some previous studies had found (we count 2 independent studies, Poort et al, 2015 and Henschke et al 2020). However, there are other studies which did not find differences (Failor et al, 2021, and Corbo et al, 2022 and ours). It is beyond the goal of our paper to reconcile these studies, but we add a point of discussion for

this (Lines 434-440). When the reviewer refers to “numerous” studies, we assume they are referring to the studies cited in our introduction, most of which look at changes in tuning rather than changes in the number of selective neurons.

2) We think our study stands on its own, not just from its relation to previous studies. This is because unsupervised representation learning has long been postulated by theorists and experimentalists alike, but relatively little evidence for it has existed so far. We also know from current deep learning research that unsupervised learning is a great way to build representations for subsequent supervised learning.

A major distinction made is that “the medial population did not represent trial-to-trial variability in mouse decision-making, while anterior HVAs did”. However, the difference between the learning and unsupervised group is the key measure here. This is the same between the medial and anterior areas.

It was not explicitly quantified, but Figure 5f did show a big difference between the supervised and unsupervised groups, and this key difference was restricted to the anterior areas. We added an explicit quantification between supervised and unsupervised areas in Figure 5g, and added quantifications for all regions in Figure S6.

Other claims include identifying a novelty response in V1, but as they mentioned themselves, this is not a novel finding.

We previously seemed to imply that our characterization of the novelty responses has been previously described in its entirety. This was a mischaracterization, it had mainly been described for passive stimulus presentations in V1. Thus, the distribution of the novelty response across brain areas is new, and the similarity of the novelty response between the supervised and unsupervised cohorts is also new. We also show how repeated exposure to a novel stimulus results in orthogonalization of neural representations, which had previously only been shown in task-trained animals. In summary, ours is a broader treatment of a previously described effect, which follows the same principle we observed for all types of plasticity: it does not require task learning. We added some text changes to emphasize and clarify this (Lines 204-206).

The improved learning rate after pre-exposure is also interesting but not novel.

The improved learning rate after pre-exposure is new in relation to the neural changes we observed. There could have been unsupervised neural plasticity without a behavioral impact, which would have been of less interest. Certainly our characterization is substantially more well-controlled than was the Tolman study from the 1920s, and this is what allows us to do a direct comparison to neural activity.

Minor: There is some phrasing which causes confusion regarding stimulus responses vs percentage of cells above a threshold of selectivity. Eg. “The responses of leaf2-selective

neurons adapted substantially after an additional week of training with the leaf2 stimulus (Figure 3a,b).” It will be important to revise all wording about “responses”, and maybe even include more analysis about response amplitudes (to the same visual stimulus) beyond the existing.

Thank you for pointing this out, we agree the wording was confusing in that paragraph and we fixed it. The coding direction analyses do include information about amplitudes of the signals, so we did not add any new analyses for that.

Referees' comments:

Referee #1 (Remarks to the Author):

The authors have addressed all of my concerns.

Referee #2 (Remarks to the Author):

The authors have done an excellent job of addressing my concerns, and I believe the updated paper should be accepted for publication.

I still have a quibble with the authors' use of the term "supervised" for their task with reward, and I don't agree with them that reinforcement learning implies a lack of perceptual learning. As the authors noted, many RL models do involve perceptual learning. In fact almost **all** modern RL models in AI involve learning sensory representations. Only toy models do not.

Added to this, I don't think the animals actually receive a supervisory signal simply because they are being asked to engage in a simple behavior (licking). Importantly, the animals do not receive a literal signal of which neurons to activate at which time (which is what is typically what is meant by "supervised" learning, at least in AI).

But, such a semantic quibble is not something that should prevent the paper from being published. I will simply leave these comments here for the authors to consider in their final revision.

Thanks. In light of this comment, we decided to rename "supervised mice" to "task mice" everywhere. This more neutral naming convention will more clearly distinguish between what the mice are trained to do, and what our interpretation is of their learning algorithms.

Referee #3 (Remarks to the Author):

While the manuscript is substantially improved in certain aspects, I remain sceptical of some key issues.

Overall, I think this manuscript does provide useful insights into the question of unsupervised exposure and how this allows subsequent improvements in supervised learning. However, I still have a concern about the generalisability of the unsupervised results, see below, and also, as mentioned by the authors and other reviewers, it is not entirely novel that prior exposure improves learning, nor that it changes neural representations, but this study puts the two together. While this is indeed an important contribution, and the orthogonalization is an interesting result, in my opinion this would not on its own be suitable for this journal.

The finding that *unsupervised exposure changes representations (in sensory cortex)* is in fact a novel result for nearly all the different conditions we described here. Based on the previous round of comments, the reviewer seems to be referring to the “mismatch response” to novel stimuli as not novel. However, that result is just one of the conditions we described (two panels in Figure 3), and even that result is novel for all brain regions except for V1.

Which is why this study goes further and makes key claims related to the comparison of unsupervised to supervised learning. Here is where I am still not convinced that the data is supporting the claims.

I have 3 broad concerns remaining:

1. I still don't think the authors are necessarily studying the relevant learning-related changes in neural responses. The authors point out their experiment with swapping the corridor in half. While this is good evidence that the mice do associate more than the start of the corridor with reward, it is still the case that the mice will most prominently learn the first view of the rewarded corridor, following which the reward is guaranteed (the lick rasters and response profiles should ideally be extended to earlier time points than 0 to allow readers to get a good sense of how early they start licking/responding).

The reviewer accepts that the swapped corridor experiment is good evidence, but then states that “it is still the case” that their opinion holds. It is ok to sometimes prioritize one's own common sense over weak evidence, but our evidence is not weak. How can the mice do the task in the swapped corridor condition if they only learned the first part of the corridor, which is now shown at the end, after the reward has passed? We have extended the time points to before 0 as the reviewer requested (Figure S6e, almost no licks before 0) and added a more granular analysis to show that mice lick in anticipation in the swapped corridors just as much as they do in the non-swapped corridor (Figure S6fg, Lines 290-294). Note also that the early licks do not lead to rewards. The reward is at a random position in the corridor, and in most mice this requires licking after the sound cue.

I don't believe it is clear what the difference is between this first-view-learning and subsequent exposure while the animal is licking and waiting for its guaranteed reward.

To reiterate, reward is not guaranteed based on early licks, which are ignored. We have also added an analysis to look at differences in neural activity between trials with early licks and late licks (Figure S4abc, Lines 189-195). The only region with substantial differences was the anterior region, which matches our previous analyses of this region having an anticipatory reward response.

There is the additional confound that the earlier parts of the corridor will have many more bouts of pre-reward licking, vs later corridor parts. Overall, while the author's claim is indeed *plausible*, I would not consider it *strong evidence* that they have characterised the relevant learning related neural changes, and thus have made a fair comparison with the unsupervised cohort.

We had previously done lick-related analyses for Figure 5, and we decided to additionally extend them in this revision. We had previously shown that “leaf2” trials had similar neural representations in the medial region whether or not there were licks (Fig 5l). In contrast, there were different representations in the anterior area in trials with and without licks (Fig 5k), and this neural response preceded the licks by up to 4 seconds (Fig 5j). We now added the other areas to this analysis (V1, lateral) to show that these areas also did not have a difference in leaf2 trials with and without licks (Figure S4d).

There are almost no leaf1 trials without licks, so we cannot do a similar analysis there. Instead, we divided leaf1 trials by early vs late licks (Figure S4a). We only saw a difference in the anterior region, but not in the medial region, where the majority of unsupervised plasticity happened (Figure S4bc).

2. Confounding responses to overt behaviour: The authors respond that since they find no difference between learning and unsupervised cohorts, this means concerns about contamination of responses with overt behaviour is not a concern. I agree that this is also a plausible argument, but I would argue that unless behavioural responses are not dealt with carefully, we are essentially dealing with poorer quality data, and there may be unexpected effects cancelling each other out. Perhaps the currently visible but subtle pattern of differences within a brain region between the cohorts (eg Fig 1i) will get enhanced, or perhaps the claim of the authors is made even stronger. Again, the claim as it stands is plausible but not the highest quality of data expected.

Our initial response to this concern was multi-pronged. Yes, one of our arguments was that we did not see differences in neural representations *despite* differences in behavior. But we also had done new behavior analyses (Figure S2) and pointed to existing behavior analyses (Figure 5). To further address such behavior confounds, we added additional analyses in this revision (Figure S4 abc, Figure S4d, Figure S6fg, Lines 189-195, 290-294, 362). Altogether, these results point to a lack of interaction between licking behavior and neural representations, except possibly in the anterior region. Even there though, a different analysis (Figure 5ij) shows that the neural ramping signal precedes the first-lick in the corridor by up to 4 seconds.

3. Taking the current results at face value, there are still some inconsistencies. First, a previous point of mine was misunderstood about the differences between supervised and unsupervised cohorts. I was referring to Fig 1j, ‘medial’ category, post learning, the data is as different as in the ‘anterior’ category. This suggests that medial and anterior both show a stronger effect of supervised compared to unsupervised (could be tested more directly by testing for an interaction effect on the pair of lines). Thus, there are more differences between unsupervised and supervised than suggested.

Sorry about the confusion. The specific comment from the first round of review was: *A major distinction made is that “the medial population did not represent trial-to-trial variability in mouse decision-making, while anterior HVAs did”*. This is a sentence in the discussion which was

immediately followed by a clear reference "(Figure 5k,l)", so we understood the comment as referring to this analysis.

More generally, we are not able to directly compare the supervised and unsupervised groups for the imaging mice due to sample size limitations. Our sample sizes are 4-5 for the supervised group, and 9 for the unsupervised group. This is generally fine for the paired before/after comparisons (higher statistical power from paired tests) where we are looking at 2-5x fold differences. The reviewer is pointing to one possible quantitative difference (Fig 1j, medial region, task mice vs unsupervised mice) that would be on the order of a 20-30% difference. With the kind of variability we see across mice (+/- 25%), a sample size calculator indicates we would need at least 10 mice per group to detect a difference. Thus, it is uncertain whether a quantitative difference really exists between the two cohorts. Even if it did, our conclusions are mostly about the qualitative similarities: yes, task learning drives plasticity in the medial area, but it also does so with unsupervised exposure. We leave it to future work to tease apart potential quantitative differences with larger sample size studies.

The reason we pointed out the difference in the anterior region for this panel (1j) was that it is a qualitative difference, not just a quantitative one. There seems to be no plasticity at all in the unsupervised cohort. We were also anticipating the full blown analyses of Figure 5, where we show that the response patterns in the anterior region are actually quite different, and come out much more clearly when selectivity is measured based on the ramping response.

Finally, we added the word "mostly" to the sentence below at the end of the first results section (Line 129). This omission was an oversight, since we had clearly described a difference in the anterior area even at the early stage of Figure 1:

*Thus, the distribution of neural plasticity across visual regions *mostly* did not depend on task feedback or supervision.*

Second, the new results with gratings, Supp Fig 1d, shows almost no changes before vs after exposure, in contrast to Fig 1i with natural stimuli. This raises doubts about how general the findings of the unsupervised exposure are.

Sorry about the misunderstanding for this analysis. The grating experiments, as requested by reviewer 2, were meant to show that there is no change in the representation of the *natural images* when gratings are presented over multiple weeks instead. We clarified this in the text (Lines 76-78, 109).